# Plant-associated $CO_2$ mediates long-distance host location and foraging behaviour of a root herbivore

**Carla CM Arce[1†], Vanitha Theepan[2†], Bernardus CJ Schimmel[2], Geoffrey Jaffuel[1], Matthias Erb[2]\*, Ricardo AR Machado[1,2]\***

[1]Institute of Biology, University of Neuchâtel, Neuchâtel, Switzerland; [2]Institute of Plant Sciences, University of Bern, Bern, Switzerland

**Abstract** Insect herbivores use different cues to locate host plants. The importance of $CO_2$ in this context is not well understood. We manipulated $CO_2$ perception in western corn rootworm (WCR) larvae through RNAi and studied how $CO_2$ perception impacts their interaction with their host plant. The expression of a carbon dioxide receptor, *DvvGr2*, is specifically required for dose-dependent larval responses to $CO_2$. Silencing $CO_2$ perception or scrubbing plant-associated $CO_2$ has no effect on the ability of WCR larvae to locate host plants at short distances (<9 cm), but impairs host location at greater distances. WCR larvae preferentially orient and prefer plants that grow in well-fertilized soils compared to plants that grow in nutrient-poor soils, a behaviour that has direct consequences for larval growth and depends on the ability of the larvae to perceive root-emitted $CO_2$. This study unravels how $CO_2$ can mediate plant–herbivore interactions by serving as a distance-dependent host location cue.

**\*For correspondence:**
matthias.erb@ips.unibe.ch (ME);
ricardo.machado@unine.ch
(RARM)

[†]These authors contributed equally to this work

**Competing interests:** The authors declare that no competing interests exist.

## Introduction

Insect herbivores can use different cues to locate suitable host plants from a distance. Volatile cues, in particular, can convey information about the identity and physiological status of a host plant and are integrated by herbivores to locate host plants for oviposition and feeding (*Visser and Avé, 1978*). Over the years, many attractive and repellent plant volatiles were identified (*Bruce et al., 2005*; *Späthe et al., 2013*; *Webster and Cardé, 2017*), and the importance of individual compounds and volatile blends was documented using synthetic chemicals (*Bruce and Pickett, 2011*; *Carrasco et al., 2015*; *Fraenkel, 1959*; *Dorn et al., 2003*; *Visser and Avé, 1978*). More recently, molecular manipulative approaches were used to manipulate plant volatile production and herbivore perception in vivo (*Fandino et al., 2019*; *Halitschke et al., 2008*; *Robert et al., 2013*), thus confirming the important role of plant volatiles in plant–herbivore interactions.

While the role of plant volatiles such as green-leaf volatiles, aromatic compounds, and terpenes is well understood, much less is known about the role of plant-associated carbon dioxide ($CO_2$) in plant–herbivore interactions. As many plant organs and their associated microbial communities release $CO_2$, it may be integrated into herbivore foraging as a marker of metabolic activity. *Datura wrightii* flowers, for instance, emit the highest levels of $CO_2$ during times of high nectar availability; as hawkmoth pollinators are attracted to $CO_2$, they may thus use this cue to locate rewarding flowers (*Goyret et al., 2008*; *Guerenstein et al., 2004*; *Guerenstein and Hildebrand, 2008*; *Stange, 1996*; *Stange, 1999*; *Stange and Stowe, 1999*; *Thom et al., 2004*). Similarly, lesions in apples result in high $CO_2$ release and attract *Bactrocera tryoni* fruit flies. As $CO_2$ at corresponding concentrations is attractive to the flies, it has been suggested that they may use plant-associated $CO_2$ to locate suitable oviposition sites (*Stange, 1999*).

**eLife digest** Living deep in the ground and surrounded by darkness, soil insects must rely on the chemicals released by plants to find the roots they feed on. Carbon dioxide, for example, is a by-product of plant respiration, which, above ground, is thought to attract moths to flowers and flies to apples; underground, however, its role is still unclear. This gaseous compound can travel through soil and potentially act as a compass for root-eating insects. Yet, it is also produced by decaying plants or animals, which are not edible. It is therefore possible that insects use this signal as a long-range cue to orient themselves, but then switch to another chemical when closer to their target to narrow in on an actual food source.

To test this idea, Arce et al. investigated whether carbon dioxide guides the larvae of Western corn rootworm to maize roots. First, the rootworm genes responsible for sensing carbon dioxide were identified and switched off, making the larvae unable to detect this gas. When the genetically engineered rootworms were further than 9cm from maize roots, they were less able to locate that food source; closer to the roots, however, the insects could orient themselves towards the plant. This suggests that the insects use carbon dioxide at long distances but rely on another chemicals to narrow down their search at close range.

To confirm this finding, Arce et al. tried absorbing the carbon dioxide using soda lime, leading to similar effects: carbon dioxide sensitive insects stopped detecting the roots at long but not short distances. Additional experiments then revealed that the compound could help insects find the best roots to feed on. Indeed, eating plants that grow on rich terrain – for instance, fertilized soils – helps insects to grow bigger and faster. These roots also release more carbon dioxide, in turn attracting rootworms more frequently.

In the United States and Eastern Europe, Western corn rootworms inflict major damage to crops, highlighting the need to understand and manage the link between fertilization regimes, carbon dioxide release and how these pests find their food.

Root-feeding insects are highly attracted to $CO_2$ in vitro (*Bernklau and Bjostad, 1998a*; *Bernklau and Bjostad, 1998b*; *Eilers et al., 2012*; *Hibbard and Bjostad, 1988*; *Jones and Coaker, 1978*; *Klingler, 1966*; *Nicolas and Sillans, 1989*; *Rogers et al., 2013*; *Strnad et al., 1986*; *Strnad and Dunn, 1990*). Given that $CO_2$ is produced and released by plant roots and diffuses relatively well through the soil, a likely explanation for this phenomenon is that root herbivores use $CO_2$ as a host location cue (*Bernklau and Bjostad, 1998a*; *Bernklau and Bjostad, 1998b*; *Doane et al., 1975*; *Erb et al., 2013*; *Johnson and Gregory, 2006*; *Johnson and Nielsen, 2012*), However, the reliability of $CO_2$ as a host location cue for root feeders has been questioned due to a number of reasons: (i) $CO_2$ can be emitted by many other sources apart from host plant roots, including decaying organic matter, microorganisms, and non-host plants; (ii) there is a strong diurnal fluctuation in plant $CO_2$ emissions that does not necessarily match with insect foraging habits; and (iii) other plant-released chemicals can be used by root herbivores for host location within a $CO_2$ background (*Agus et al., 2010*; *Eilers et al., 2012*; *Erb et al., 2013*; *Hansen, 1977*; *Hibbard and Bjostad, 1988*; *Hiltpold and Turlings, 2012*; *Johnson and Nielsen, 2012*; *Reinecke et al., 2008*; *Weissteiner et al., 2012*). A model that may reconcile these different views is that $CO_2$ may be used as an initial cue at long distances, while other, more host-specific volatiles may be used at shorter distances (*Erb et al., 2013*; *Johnson et al., 2006*; *Johnson and Nielsen, 2012*). So far, this model has not been experimentally validated, and the precise role of plant-associated $CO_2$ as a host location cue by herbivores, in general, and root herbivores, in particular, remains unclear (*Eilers et al., 2016*). To the best of our knowledge, no studies so far have investigated the role of plant-associated $CO_2$ in plant–herbivore interactions in vivo using molecular manipulative approaches.

The larvae of *Diabrotica virgifera virgifera* (the western corn rootworm [WCR]) feed almost exclusively on maize roots in agricultural settings and cause major yield losses in the US and Eastern Europe (*Ciosi et al., 2008*; *Gray et al., 2009*; *Meinke et al., 2009*; *Toepfer et al., 2015*). The larvae rely on a number of volatile and non-volatile chemicals to identify and locate host plants, and distinguish between suitable and less-suitable maize plants and forage within the maize root system (*Hiltpold et al., 2013*; *Johnson and Gregory, 2006*; *Johnson and Nielsen, 2012*;

*Robert et al., 2012c*; *Schumann et al., 2018*). Non-volatile primary metabolites such as sugars and fatty acids as well as secondary metabolites such as benzoxazinoids and phenolic acid conjugates modulate larval behaviour (*Bernklau et al., 2011*; *Bernklau et al., 2015*; *Bernklau et al., 2016a*; *Bernklau et al., 2018*; *Erb et al., 2015*; *Hu et al., 2018*; *Huang et al., 2017*; *Machado et al., 2021*; *Robert et al., 2012c*). Volatiles including (*E*)-β-caryophyllene, ethylene, and $CO_2$ attract the larvae (*Bernklau and Bjostad, 1998a*; *Bernklau and Bjostad, 1998b*; *Robert et al., 2012b*; *Robert et al., 2012a*), while methyl anthranilate repels them (*Bernklau et al., 2016b*). Based on the finding that high $CO_2$ levels can outweigh the attractive effects of other maize volatiles, it was suggested that $CO_2$ may be the only relevant volatile attractant for WCR larvae (*Bernklau and Bjostad, 1998b*). However, under conditions where $CO_2$ levels are similar, WCR larvae reliably choose between host plants of different suitability using other volatile cues (*Huang et al., 2017*; *Lu et al., 2016*; *Robert et al., 2012b*; *Robert et al., 2012a*). The demonstrated ability of WCR larvae to respond to different volatile cues and the recent identification of putative $CO_2$ receptors from transcriptomic data (*Rodrigues et al., 2016*) make this species a suitable model system to investigate the role of $CO_2$ in plant–herbivore interactions. Ongoing efforts to use $CO_2$ as a bait to control WCR in the field (*Bernklau et al., 2004*; *Schumann et al., 2014a*; *Schumann et al., 2014b*) provide further motivation to assess the importance of this volatile for WCR foraging.

To understand the importance of $CO_2$ for WCR foraging in the soil, we manipulated the insect's capacity to perceive $CO_2$. We reduced the expression levels of three putative WCR $CO_2$ receptor-encoding genes through RNA interference (RNAi), resulting in the identification of *DvvGr2* as an essential gene for $CO_2$ perception. Using *DvvGr2*-silenced larvae in combination with $CO_2$ removal, we then assessed the importance of $CO_2$ perception for WCR behaviour and foraging in olfactometers and soil arenas. Our experiments reveal how root-associated $CO_2$ modulates the interaction between maize and its economically most damaging root pest and expand the current repertoire of potential adaptive explanations for the attraction of insect herbivores to $CO_2$.

## Results

### Plants create $CO_2$ gradients in the soil

Plant-emitted $CO_2$ may be used as a host location cue by root herbivores. To understand whether the presence of plant roots is associated with higher $CO_2$ levels, we measured $CO_2$ levels in the soil at different distances from young maize seedlings. We observed a significant $CO_2$ gradient in the soil, with concentrations of 548–554 ppm in the rhizosphere, 506–515 ppm at distances between 8 and 16 cm from the plant, and 460–484 ppm at distances between 16 and 32 cm (*Figure 1A*). At distances of more than 40 cm, $CO_2$ levels levelled off at 425–439 ppm. We then removed the plants and remeasured $CO_2$ levels 1 hr afterwards. In the absence of the plants, no $CO_2$ gradient was observed, and $CO_2$ concentrations in the soil were around 430 ppm (*Figure 1B*). When surrounding soil was removed and seedling roots were washed, we observed $542 \pm 6.74$ ppm $CO_2$ around the roots (n = 3). Thus, the release of $CO_2$ from maize roots can account for the $CO_2$ difference between soil trays with and without plants. This experiment shows that elevated $CO_2$ levels derived from roots and probably from root-associated microorganisms are temporally and spatially associated with the presence of maize roots, and may thus be used as a host location cue by the WCR. To test this hypothesis, we identified $CO_2$ receptors in WCR larvae, genetically impair their expression, and conducted a series of behavioural experiments as described below.

### The WCR genome encodes three putative $CO_2$ receptors

To identify genes encoding putative $CO_2$ receptors in WCR, we used known $CO_2$ receptor-encoding gene sequences as queries against the WCR genome (available from the National Center for Biotechnology Information [NCBI]). Three putative carbon dioxide receptor candidates, *DvvGr1*, *DvvGr2*, and *DvvGr3*, were identified, matching three candidate genes that were found in previous transcriptome analyses (*Rodrigues et al., 2016*). Phylogenetic reconstruction based on in silico-predicted protein sequences revealed orthologous relationships for the three WCR candidate receptors and the receptors of several other insects (*Figure 2A*). Consistent with their taxonomy, we observed close homology between the protein sequences of the $CO_2$ receptors of WCR and the protein sequences of other coleopteran insects such as *Tribolium castaneum* (*Figure 2A*). Expression levels

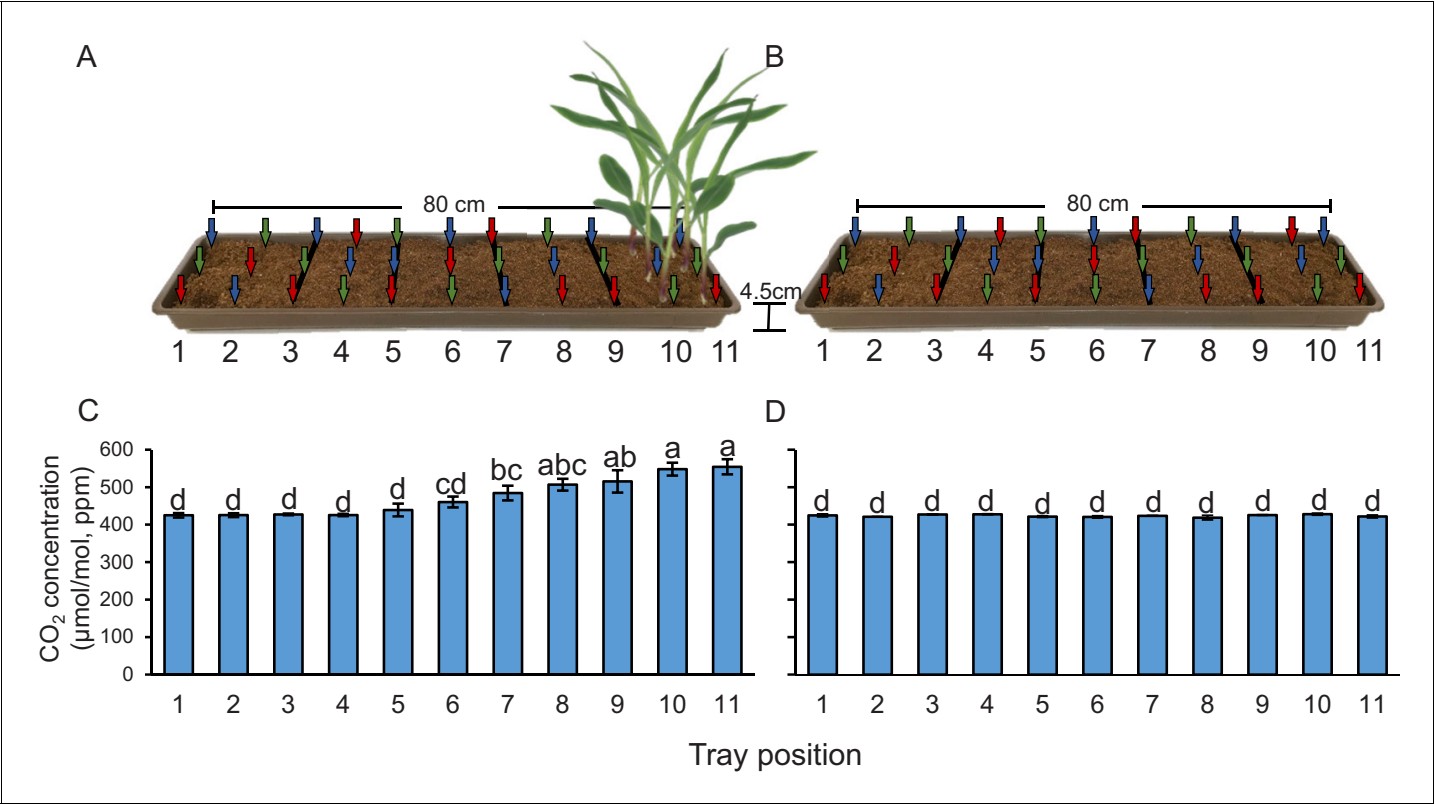

**Figure 1.** Plants create $CO_2$ gradients in the soil. (A, B) $CO_2$ levels were determined in soil-filled trays at different distances from young maize seedlings (A) before and (B) after removing the seedlings from the system. Arrows indicate air sampling points. Different colours indicate sampling positions within three individual trays that were assayed (n = 3). Red arrows indicate samplings points in tray 1, green arrows indicate samplings points in tray 2, and blue arrows indicate samplings points in tray 3. (C, D) Mean (± SEM) $CO_2$ levels at different distances from the plant (C) before and (D) after removing the seedlings from the system. Different letters indicate significant differences in $CO_2$ levels in each tray position (p < 0.05 by two-way ANOVA with Holm's multiple-comparisons test). For details regarding the statistical results, refer to *Supplementary file 1*. Raw data are available in *Figure 1—source data 1*.

The online version of this article includes the following source data for figure 1:

**Source data 1.** Raw data for *Figure 1*.

of *DvvGr1* and *DvvGr2* were found to be significantly higher in the head than in the rest of the body (thorax and abdomen) of second instar WCR larvae (*Figure 2B, C*). No significant difference in expression was observed for *DvvGr3* (*Figure 2D*). Protein tertiary structure and topology models indicated that all three genes encode for 7-transmembrane domain proteins, which is consistent with their roles as receptors (*Figure 2B–D*).

### *DvvGr2* expression is specifically required for responsiveness of WCR larvae to $CO_2$

To determine the importance of *DvvGr1*, *DvvGr2*, and *DvvGr3* for the responsiveness of WCR larvae to $CO_2$, we knocked down the expression of each gene individually through double-stranded RNA (dsRNA)-mediated RNAi and conducted initial behavioural experiments with carbonated water as a $CO_2$ source (*Figure 3*). Oral administration of dsRNA targeting either *DvvGr1*, *DvvGr2*, or *DvvGr3* reduced the expression levels of these genes by 80%, 83%, and 66% compared to WCR larvae fed with dsRNA of the green fluorescent protein (GFP) gene (herein referred to as wild type [WT]) (*Figure 3A*). All RNAi constructs were confirmed to be gene specific (*Figure 3A*). Measurements within the olfactometers showed that $CO_2$ levels were approximately 100 ppm higher in the arms of the L-shaped pots that contained plastic cups filled with carbonated water than in the arms of L-shaped pots that contained plastic cups filled with distilled water (*Figure 3B*, *Figure 3—figure supplement 1*). A higher proportion of WT larvae moved towards olfactometer arms with higher

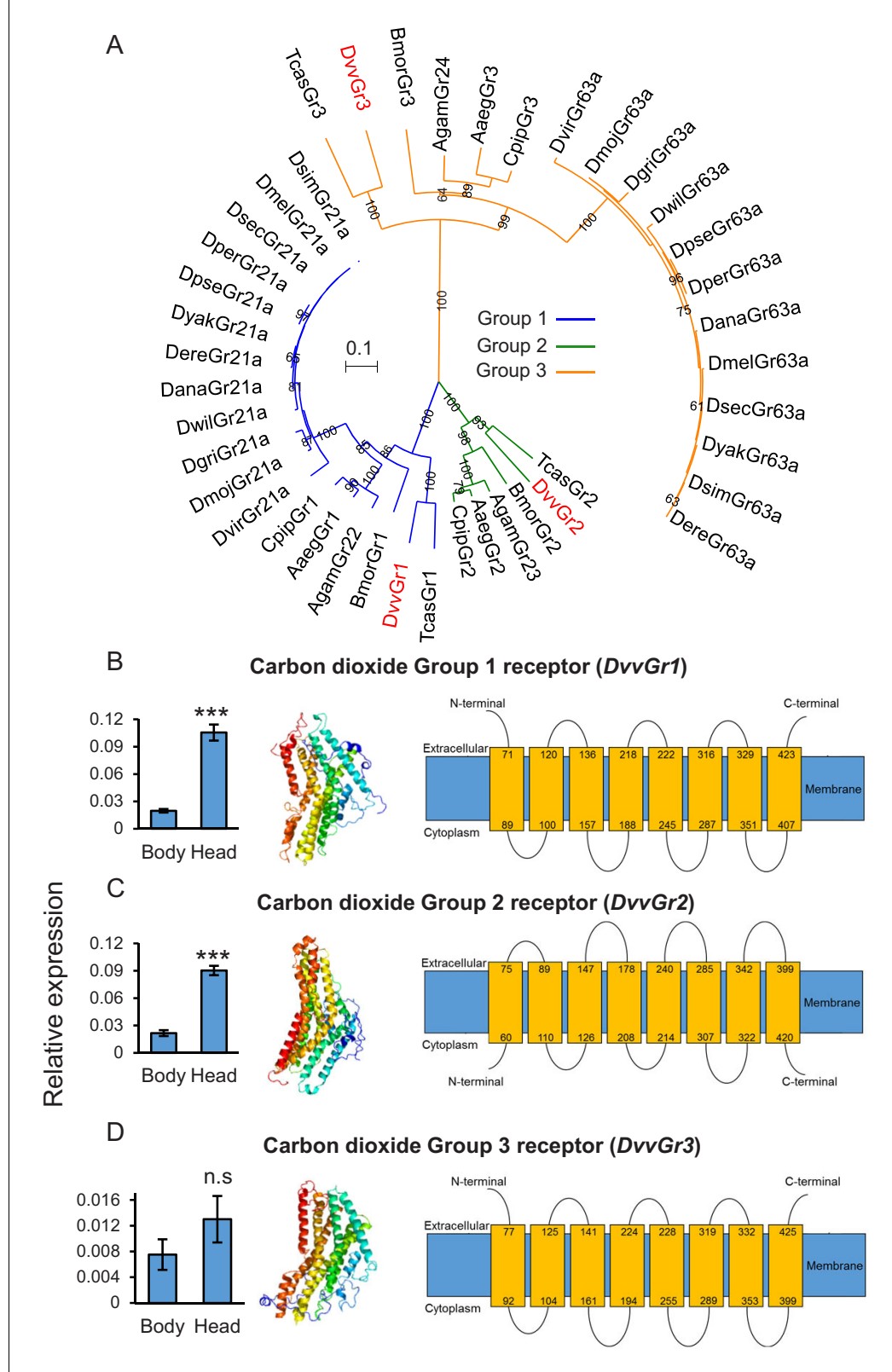

**Figure 2.** The western corn rootworm (WCR) genome contains three putative carbon dioxide ($CO_2$) receptors. (**A**) Phylogenetic relationships between putative $CO_2$ receptors based on protein sequences of different insects. Dmel: *Drosophila melanogaster*; Dsim: *Drosophila simulans*; Dsec: *Drosophila sechellia*; Dyak: *Drosophila yakuba*; Dere: *Drosophila erecta*; Dana: *Drosophila ananassae*; Dper: *Drosophila persimilis*; Dpse: *Drosophila pseudoobscura*; *Figure 2 continued on next page*

*Figure 2 continued*

Dwil: *Drosophila willistoni*; Dgri: *Drosophila grimshawi*; Dmoj: *Drosophila mojavensis*; Dvir: *Drosophila virilis*; Agam: *Aedes gambiae*; Aaeg: *Aedes aegypti*; Cqui: *Culex quinquefasciatus*; Bmor: *Bombyx mori*; Tcas: *Tribolium castaneum*; Dvv: *Diabrotica virgifera virgifera* (WCR). Evolutionary relationships were inferred using the neighbor-joining method. The optimal tree with the sum of branch length = 4.44068889 is shown. The percentage of replicate trees in which the associated taxa clustered together in the bootstrap test (100 replicates) are shown next to the branches. The tree is drawn to scale, with branch lengths in the same units as those of the evolutionary distances used to infer the phylogenetic tree. The evolutionary distances were computed using the Poisson correction method and are in the units of the number of amino acid substitutions per site. A total of 242 amino acid positions were included in the final data set. (B–D) Mean (± SEM) relative gene expression levels of group 1 (*DvvGr1*) (B), group 2 (*DvvGr2*) (C), and group 3 (*DvvGr3*) (D) $CO_2$ receptors in the bodies (thorax and abdomen) or heads of second instar WCR larvae (n = 10). Asterisks indicate statistically significant differences between tissue types within genes (\*\*\*p < 0.001 by Student's *t* test; n.s.: not significant). For details regarding the statistical results, refer to *Supplementary file 1*. Raw data are available in *Figure 2—source data 1*. (B–D) Predicted protein tertiary structure (left) and transmembrane protein topology (right) of (B) *DvvGr1*; (C) *DvvGr2*, and (D) *DvvGr3* according to the Phyre2 algorithm.

The online version of this article includes the following source data for figure 2:

**Source data 1.** Raw data for *Figure 2*.

$CO_2$ levels (*Figure 3C*). Silencing *DvvGr1* or *DvvGr3* expression did not alter this preference. In contrast, *DvvGr2*-silenced larvae did not show preference for any olfactometer arm (*Figure 3C*). To explore the role of *DvvGr2* in different aspects of WCR behaviour, we conducted a series of additional experiments. First, we assessed the impact of silencing *DvvGr2* on the capacity of WCR larvae to respond to other volatile and non-volatile host cues (*Figure 3D–G*). *DvvGr2*-silenced larvae responded similarly to the repellent volatile methyl anthranilate as WT larvae (*Figure 3E*). Responsiveness to non-volatile compounds such as Fe(III)(DIMBOA)$_3$ and a blend of glucose, fructose, and sucrose was also unaltered in *DvvGr2*-silenced larvae (*Figure 3F, G*), demonstrating that knocking down *DvvGr2* expression does not alter the capacity of WCR larvae to respond to other important chemical cues. Second, we assessed the contribution of *DvvGr2* to $CO_2$ responsiveness using synthetic $CO_2$ at different concentrations (*Figure 4*, *Figure 4—figure supplement 1*). WT larvae showed characteristic dose-dependent behavioural responses to $CO_2$. While they did not respond to 22 ppm $CO_2$ above ambient $CO_2$ levels, they were attracted to $CO_2$ concentrations between 59 and 258 ppm above ambient and repelled by $CO_2$-enriched air at 950 ppm above ambient $CO_2$ levels and above (*Figure 4*). In contrast, *DvvGr2*-silenced larvae did not respond to $CO_2$ enrichment at any of the tested concentrations (*Figure 4*). These experiments show that WCR larvae are attracted to $CO_2$-enriched environments within the physiological range of the maize rhizosphere and that *DvvGr2* silencing fully and specifically suppresses $CO_2$ responsiveness in WCR larvae.

## *DvvGr2* expression does not affect larval motility or short-range host location

To assess the impact of *DvvGr2* on larval motility, we followed the trajectories of individual larvae in humid filter paper-lined Petri plates that were outfitted with a $CO_2$ point releaser (*Figure 5*). WT larvae made frequent turns, but consistently oriented themselves towards the $CO_2$ release point. Once they reached the $CO_2$ release point, they stopped moving (*Figure 5A*). *DvvGr2*-silenced larvae exhibited similar turning behaviour as WT larvae, but did not move towards the $CO_2$ release point (*Figure 5B*). WT larvae spend more time on $CO_2$ release point than *DvvGr2*-silenced larvae (*Figure 5C*). During the movement phase, the mean speed of WT larvae and *DvvGr2*-silenced larvae was similar (*Figure 5C*), but the distance covered by *DvvGr2*-silenced larvae was higher, as they did not stop at the $CO_2$ release point. In a second experiment, we followed the trajectories of individual larvae in Petri plates with maize roots (*Figure 5D, E*). Speed and distance covered were similar between WT and *DvvGr2*-silenced larvae (*Figure 5F*). Surprisingly, both WT and *DvvGr2*-silenced larvae oriented themselves towards the maize roots and reached the maize roots after a similar amount of time (*Figure 5F*). This result shows that *DvvGr2* expression is required for the location and detection of $CO_2$, but does not influence WCR motility nor its ability to locate maize roots over short distances (i.e.,<9 cm).

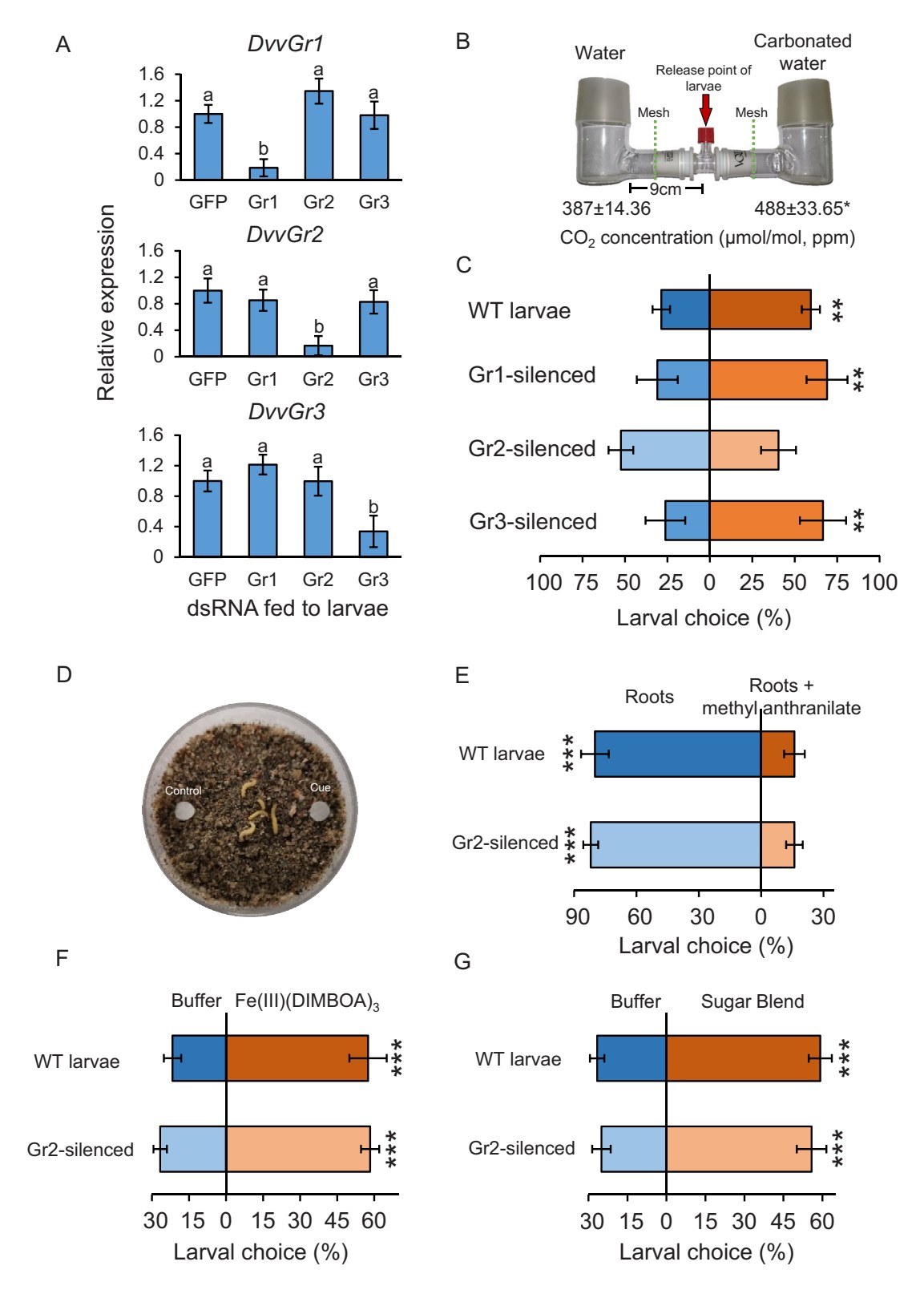

**Figure 3.** The carbon dioxide group 2 receptor (DvvGr2) is specifically required for the attraction of western corn rootworm (WCR) towards $CO_2$. (A) Mean (± SEM) relative gene expression levels of group 1 (*DvvGr1*), group 2 (*DvvGr2*), and group 3 (*DvvGr3*) $CO_2$ receptors after WCR larvae were fed with dsRNA-expressing bacteria targeting green fluorescent protein (GFP, herein referred to as WT), *DvvGr1*, *DvvGr2*, or *DvvGr3* genes (n = 11–13). Different letters indicate significant differences of gene expression levels (p < 0.05 by one-way ANOVA with Holm's multiple-comparisons test). (B) *Figure 3 continued on next page*

*Figure 3 continued*

Mean (± SEM) $CO_2$ levels in each L-shaped pot of the two-arm belowground olfactometers used to test the attractive and repellent effects of $CO_2$ to WCR larvae (n = 4–8). Asterisk indicates significant differences in $CO_2$ levels (*p < 0.05 by Student's *t* test). For detailed data on $CO_2$ levels, refer to *Figure 3—figure supplement 1*. (C) Mean (± SEM) proportion of WCR larvae observed in the olfactometer arms with higher $CO_2$ levels (carbonated water side) or in control arms (distilled water side). Larvae were considered to have made a choice when they were found at a distance of 1 cm or less from the wire mesh (indicated by dashed green lines). Seven olfactometers with six larvae each were assayed (n = 7). (D) Petri plates used to test insect responses to methyl anthranilate, to Fe(III)(DIMBOA)$_3$, and to sugars. (E) Mean (± SEM) proportion of WCR larvae observed on roots or on roots placed next to filter paper discs impregnated with methyl anthranilate, (F) on filter paper discs impregnated with buffer or with Fe(III)(DIMBOA)$_3$, and (G) on filter paper discs impregnated with buffer or with a blend of glucose, fructose, and sucrose. For (E), 10 choice arenas with five larvae each were assayed (n = 10). Larvae were considered to have made a choice when they made physical contact with the roots or the filter paper discs. For (F, G), 20 choice arenas with six larvae each were assayed (n = 20). Asterisks indicate statistically significant differences in larval choices between treatments (***p < 0.001 by generalized linear model followed by False discovery rate (FDR)-corrected post hoc tests). Note that the number of replicates across experiments varied depending on the availability of insects. For details regarding the statistical results, refer to *Supplementary file 1*. Raw data are available in *Figure 3—source data 1*.

The online version of this article includes the following source data and figure supplement(s) for figure 3:

**Source data 1.** Raw data for *Figure 3*.
**Figure supplement 1.** Carbon dioxide levels at different sampling points.

## Root-associated $CO_2$ enhances volatile-mediated host location by WCR larvae in a distance-specific manner

To further explore the role of plant-associated $CO_2$ and *DvvGr2* in volatile-mediated host location, we performed a series of olfactometer experiments with maize plants grown in sand on one side and sand only on the other side. We tested attraction at two distances, 9 and 18 cm, from the volatile sources and the release points of the larvae (*Figure 6*). We also manipulated the diffusion of $CO_2$ into the arms of a subset of olfactometers by adding a layer of $CO_2$-absorbing soda lime into the olfactometer arms. $CO_2$ measurements revealed that the presence of a host root system increased $CO_2$ concentrations by approximately 100 ppm above ambient $CO_2$ levels in the corresponding olfactometer arm (*Figure 6*, *Figure 6—figure supplement 1*). The soda lime reduced ambient $CO_2$ concentrations in the olfactometer arms by approximately 100 ppm and equalized $CO_2$ concentrations between arms with and without a host plant (*Figure 6*). The diffusion of other maize root volatiles was not affected by the soda lime (*Figure 6—figure supplement 2*), thus validating the $CO_2$ scrubbing approach. Larvae did not have direct access to the plant, the plant growth medium, or the soda lime, and received no visual cues, and thus had to rely on host plant volatiles for orientation. When released at distance of 9 cm from the volatile sources, both WT and *DvvGr2*-silenced larvae showed a clear preference for the olfactometer arms leading to host plants (*Figure 6A*). This preference was still intact in olfactometers outfitted with soda lime, showing that volatiles other than $CO_2$ are sufficient for volatile-mediated host location at a short distance. At a distance of 18 cm from the volatile sources, WT larvae showed a similarly strong preference for arms leading to host plants (*Figure 6B*). By contrast, *DvvGr2*-silenced larvae did not exhibit any preference (*Figure 6B*). In the presence of soda lime, neither WT nor *DvvGr2*-silenced larvae were attracted to arms with a host plant (*Figure 6B*). Taken together, these experiments provide strong support for the hypothesis that WCR larvae use plant-associated $CO_2$ to locate host plants over distances greater than 9 cm in a *DvvGr2*-dependent manner.

## Root-associated $CO_2$ enhances volatile-mediated host location by WCR larvae in a distance-specific manner

WCR larvae can move up to 1 m in the soil. Second and third instar larvae in particular are known to move between maize plants across rows in maize fields (*Hibbard et al., 2003*). To test whether *DvvGr2*-mediated $CO_2$ responsiveness mediates host location over longer distances in a soil context, we planted maize plants in soil-filled plastic trays, released WCR larvae at distances of 16, 32, 48, or 64 cm from the maize plants, and evaluated larval positions after 8 hr (*Figure 7*). This time point was chosen based on preliminary observations showing that larvae take approximately 8 hr to cross the soil arenas. Direct access to the roots was impeded by using volatile-permeable fabrics, referred to hereby as root barriers. The $CO_2$ emitted by maize roots formed a gradient in the soil, starting at about 506 ppm in the rhizosphere (zone 1) and 430 ppm at distances of 16–32 cm from the plant

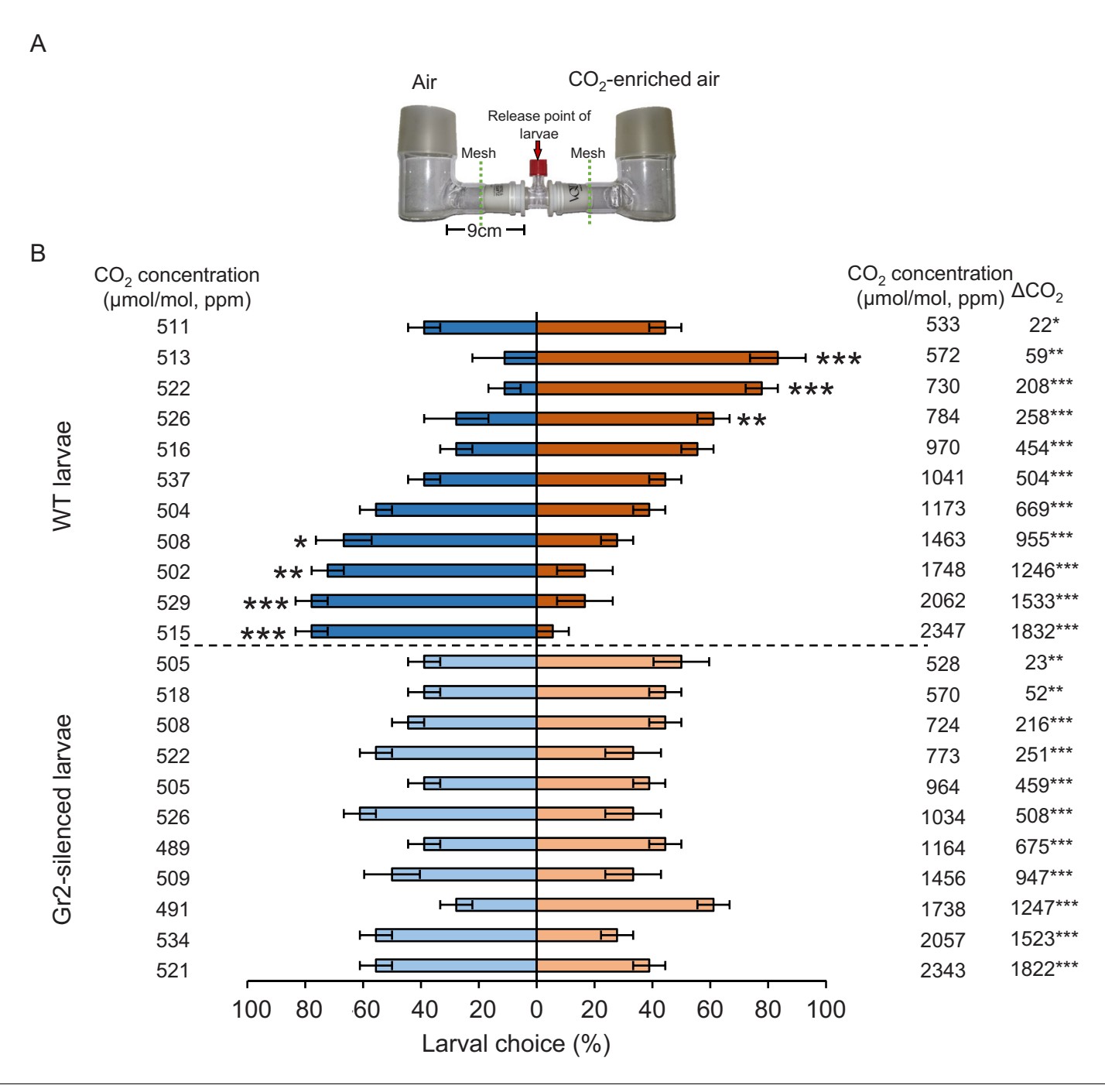

**Figure 4.** DvvGr2 is required for dose-dependent western corn rootworm (WCR) responses to $CO_2$. (**A**) Two-arm olfactometer used to test the attractive and repellent effects of $CO_2$ on WCR larvae. (**B**) Mean ( ± SEM) proportion of WCR larvae observed in each arm of the olfactometers. Larvae were considered to have made a choice when they were found at a distance of 1 cm or less from the wire mesh, indicated by dashed green lines. Three olfactometers with six larvae each were assayed (n = 3). Asterisks indicate statistically significant differences between larval choices (*p < 0.05; **p < 0.01; ***p < 0.001 by generalized linear model [GLM] followed by FDR-corrected post hoc tests). Mean $CO_2$ concentrations in each olfactometer side and the difference between them ($\Delta CO_2$) are indicated. Asterisks indicate significant differences in the $CO_2$ levels of each olfactometer arm (*p < 0.05; **p < 0.01; *** p < 0.001 by GLM followed by FDR-corrected post hoc tests). For detailed data on $CO_2$ levels, refer to *Figure 4—figure supplement 1*. For details regarding the statistical results, refer to *Supplementary file 1*. Raw data are available in *Figure 4—source data 1*. The online version of this article includes the following source data and figure supplement(s) for figure 4:

**Source data 1.** Raw data for *Figure 4*.
**Figure supplement 1.** Carbon dioxide levels at different sampling points.

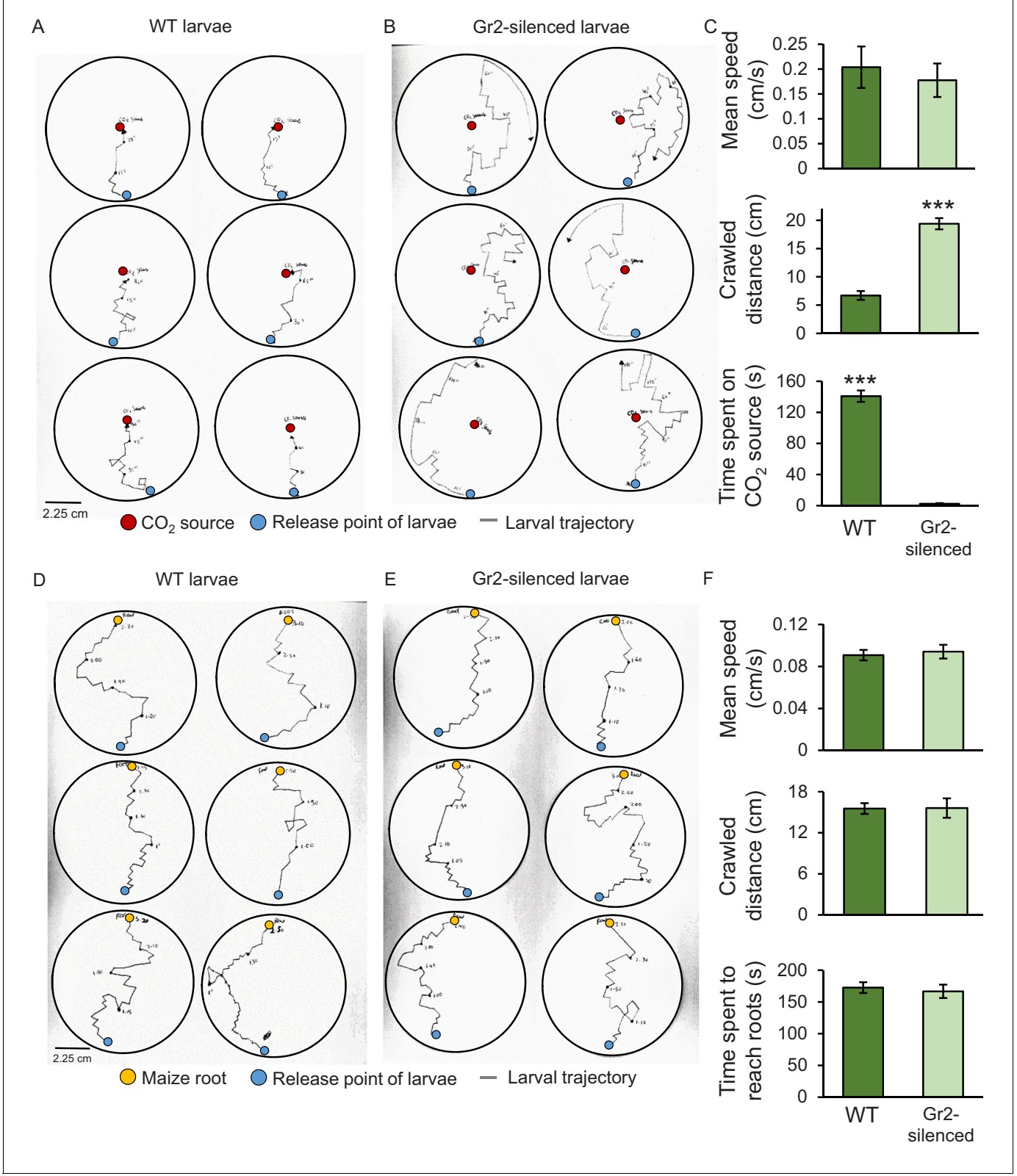

**Figure 5.** Silencing the carbon dioxide group 2 receptor (DvvGr2) impairs western corn rootworm (WCR) responses to CO₂ without affecting larval motility or search behaviour. (A, B) Trajectories of individual wild type (WT) (A) and *DvvGr2*-silenced (B) WCR larvae in Petri plates with a CO₂ source. The blue circles represent larval release points. The red circles represent CO₂ sources consisting of a fine needle that releases CO₂ at 581 ppm, resulting in CO₂ concentrations 60 ppm above ambient CO₂ levels at the release point. (C) Mean (± SEM) speed and distance covered during the
*Figure 5 continued on next page*

*Figure 5 continued*

movement phase, and time spent at the $CO_2$ source during the first 3 min of the experiment. (D, E) Trajectories followed by WT (D) and by *DvvGr2*-silenced (E) WCR larvae on Petri plates containing maize seedling roots. The blue circles represent larval release points. The yellow circles represent maize seedling roots. (F) Mean (± SEM) speed and distance covered during the movement phase, and time necessary to reach the maize root during the first 3 min of the experiment. For both experiments, six Petri plates with one larva each were assayed (n = 6). Asterisks indicate statistically significant differences between mobility parameters of WT and *DvvGr2*-silenced larvae (***p < 0.001 by Student's *t* test). For details regarding the statistical results, refer to *Supplementary file 1*. Raw data are available in *Figure 5—source data 1*.

The online version of this article includes the following source data for figure 5:

**Source data 1.** Raw data for *Figure 5*.

(zone 2) (*Figure 7B*, *Figure 7—figure supplement 1*). At distances of more than 32 cm from the plant, the $CO_2$ levels were around 400 ppm and statistically indistinguishable from soil without plants or ambient air (*Figure 7—figure supplement 1*). To confirm that larval motility is not altered by *DvvGr2* silencing in a soil context, we first released WT and *DvvGr2*-silenced larvae into the middle of a set of arenas without a host plant and evaluated larval positions after 8 hr. We found that the larvae dispersed equally across the arenas, without any difference between WT and *DvvGr2*-silenced larvae (*Figure 7A*). Eight hours after releasing the larvae into arenas that included host plants on one side, 53% of WT larvae that were released at 64 cm from the plant were retrieved close to the maize rhizosphere, that is, in zone 1 (*Figure 7C*). In contrast, only 33% of the *DvvGr2*-silenced larvae that were released at the same distance were recovered from the maize rhizosphere (*Figure 7C*). Significantly more *DvvGr2*-silenced larvae were recovered further away from the plants, in zones 3 and 4 (*Figure 7C*). The number of WT and *DvvGr2*-silenced WCR larvae found close to the host plant increased with decreasing release distance, as did the difference between WT and *DvvGr2*-silenced larvae (*Figure 7C–F*). At a release distance of 16 cm, only slightly more WT than *DvvGr2*-silenced larvae were found close to the plant roots (*Figure 7F*). To further confirm the role of *DvvGr2* in mediating host plant location over long distances in the soil, we performed a time-course experiment where we released WT and *DvvGr2*-silenced larvae in zone 5 (64 cm away from the host plant) and then recorded how rapidly they reached zone 1 containing host plants (*Figure 7—figure supplement 2*). The capacity of the larvae to directly feed on the host roots was impeded using a volatile-permeable root barrier (*Figure 7—figure supplement 2*). Within 10 hr, 36% of the released WT larvae were found in zone 1, and within 32 hr, this number had increased to 90% (*Figure 7—figure supplement 2*). By contrast, only 24% of the released *DvvGr2*-silenced larvae were found in zone 1 after 10 hr, and after 32 hr, this value had only increased to 56% (*Figure 7—figure supplement 2*). Thus, the capacity to detect $CO_2$ gradients contributes to successful host location by WCR larvae in a distance-specific manner in the soil. While larvae released at a distance equal to or below 32 cm from the host plant (zones 2–3) can use $CO_2$ directly as a host location cue, larvae released at greater distances likely move randomly before reaching zones with plant-associated $CO_2$ gradients.

## $CO_2$ perception enhances the capacity of WCR larvae to locate better hosts

Plant nutritional status determines plant growth and defence, and can thus modulate plant–herbivore interactions (*Wetzel et al., 2016*). To test for a possible connection between plant nutritional status, host suitability, and $CO_2$-dependent herbivore attraction, we varied the nutrient supply of maize plants and then carried out $CO_2$ measurements, and behavioural and insect performance experiments (*Figure 8*). To exclude direct or soil-mediated effects of fertilization, plants were first grown under different fertilization regimes and then, prior to experiments, harvested, washed, and replanted. Higher $CO_2$ levels were observed close to the roots of plants that were well fertilized compared to the levels that were observed close to the roots of plants that received medium (50% of optimally fertilized plants) or low (10% of optimally fertilized plants) fertilizer doses (*Figure 8A, B*, *Figure 8—figure supplement 1A, B*). As observed before, soil $CO_2$ levels decreased with increasing distance from the plants and were lowest in the middle of the experimental trays (*Figure 8A, B*). In choice experiments with maize plants planted approximately 50 cm apart, which corresponds to row spacing used for high planting densities in maize cultivation, WT larvae showed a significant preference for well-fertilized over medium- or low-fertilized plants (*Figure 8C, D*). *DvvGr2*-silenced larvae did not show any preference (*Figure 8C, D*). In no-choice experiments, WCR larvae gained most

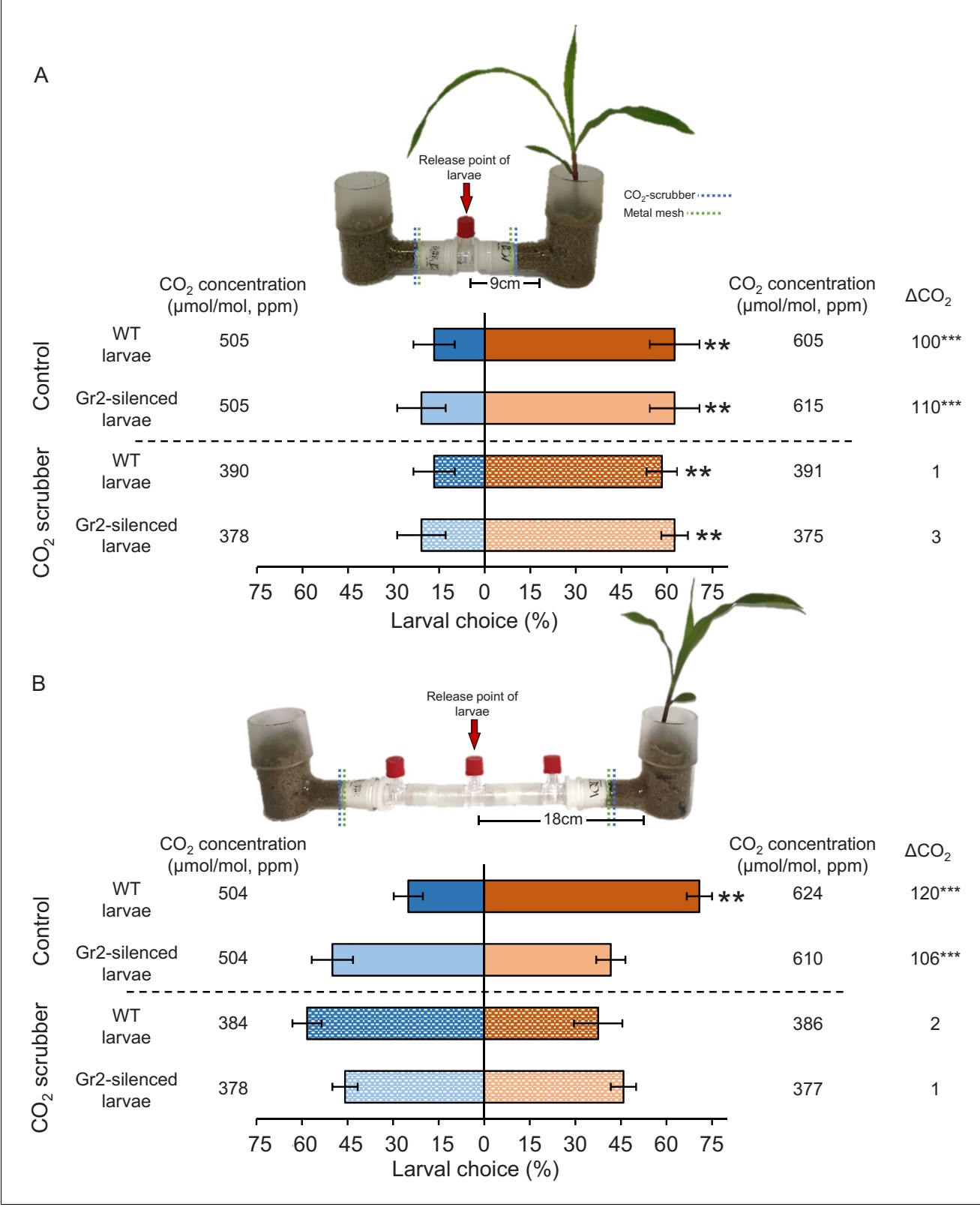

**Figure 6.** Plant-associated $CO_2$ mediates host location by western corn rootworm (WCR) larvae in a distance-specific manner. (**A, B**) Mean (± SEM) proportion (%) of WCR larvae observed on each side of the olfactometers. Larva were considered to have made a choice when they were found at a distance of 1 cm or less from the wire mesh, indicated by dashed green lines. Control olfactometers allowed for plant-associated $CO_2$ to diffuse into the central glass tubes, while $CO_2$ scrubber olfactometers were outfitted with soda lime to suppress $CO_2$ diffusion while allowing for the diffusion of

*Figure 6 continued on next page*

*Figure 6 continued*

other volatiles. Mean $CO_2$ concentrations in each olfactometer side and the difference between them ($\Delta CO_2$) are given. Asterisks indicate significant differences in the $CO_2$ levels of each olfactometer arm (\*\*\*p < 0.001 by generalized linear model [GLM] followed by FDR-corrected post hoc tests). For detailed data on $CO_2$ levels and other volatiles, refer to *Figure 6—figure supplements 1* and *2*. Four olfactometers with six larvae each were assayed using wild type (WT) or *DvvGr2*-silenced larvae (n = 4). Asterisks indicate statistically significant differences between treatments (\*\*p < 0.01 by GLM followed by FDR-corrected post hoc tests). For details regarding the statistical results, refer to *Supplementary file 1*. Raw data are available in *Figure 6—source data 1*.

The online version of this article includes the following source data and figure supplement(s) for figure 6:

**Source data 1.** Raw data for *Figure 6*.
**Figure supplement 1.** Carbon dioxide levels at different sampling points.
**Figure supplement 2.** Soda lime does not influence the diffusion of plant volatiles other than $CO_2$ into olfactometer arms.

weight on washed roots of well-fertilized maize plants than on washed roots of plants treated with medium or low doses of fertilizer (*Figure 8E*). Hence, intact $CO_2$ perception allows WCR larvae to locate suitable host plants at agriculturally relevant distances, which may result in specific insect distribution patterns in heterogeneous environments.

## Discussion

In this study, we conducted gene sequence similarity analyses, phylogenetic relationship reconstructions, RNA interference, and behavioural experiments to explore the biological relevance of root-associated $CO_2$ for plant–herbivore interactions. We found that the WCR genome contains at least three putative $CO_2$ receptor-encoding genes: *DvvGr1*, *DvvGr2*, and *DvvGr3*, which is consistent with previous transcriptomic-based studies (*Rodrigues et al., 2016*). Protein tertiary structure and topology prediction models show that the identified genes code for proteins that contain seven transmembrane domains, which is consistent with the protein topology of gustatory and olfactory receptors (*Dahanukar et al., 2005*; *Hallem et al., 2006*). Larval behaviour and gene silencing based-functional characterization of the three identified WCR putative $CO_2$ receptor genes revealed that the intact expression of *DvvGr2* is essential for the attractive effects of $CO_2$ to WCR larvae. Knocking down *DvvGr2* rendered larvae fully unresponsive to synthetic and plant-associated $CO_2$ without impairing responses to other stimuli or affecting search behaviour and motility. In *Aedes aegypti*, *Helicoverpa armigera*, and *Drosophila melanogaster*, both carbon dioxide receptors *Gr1* and *Gr3* are required for $CO_2$ detection (*Erdelyan et al., 2012*; *Jones et al., 2007*; *Kwon et al., 2007*; *McMeniman et al., 2014*; *Ning et al., 2016*; *Suh et al., 2004*). In *Culex quinquefasciatus*, both *Gr2* and *Gr3* carbon dioxide receptors are required, while *Gr1* acts as a modulator (*Xu et al., 2020*). In *A. aegypti*, the involvement of *Gr2* in carbon dioxide responsiveness is still under debate (*Erdelyan et al., 2012*; *Kumar et al., 2019*). Taken together, the molecular elements required for carbon dioxide perception may be species-specific. Our results support this notion as *DvvGr2*, but not *DvvGr1* and *DvvGr3*, are crucial for $CO_2$ responsiveness. The role of *DvvGr1* and *DvvGr3* for WCR remains to be determined, but their presence and expression may hint at additional complexity in developmental and/or tissue-specific patterns of $CO_2$ responsiveness in this species.

Despite the inability of *DvvGr2*-silenced WCR larvae to respond to differences in $CO_2$ levels, the larvae were still able to orient towards maize roots at short distances of 8–10 cm. Olfactometer experiments in combination with $CO_2$ removal demonstrate that other volatile cues can be used by WCR larvae to locate maize plants at distances shorter than 9 cm. Earlier studies found that (E)-β-caryophyllene, which is emitted from the roots of certain maize genotypes when they are attacked by root herbivores, attracts second and third instar WCR larvae and allows them to aggregate on maize plants and thereby enhance their fitness (*Robert et al., 2012b*), while neonate larvae are not attracted to this volatile (*Hiltpold and Hibbard, 2016*). Ethylene has also been shown to attract WCR larvae (*Robert et al., 2012a*), and MBOA or its breakdown products have also been proposed as volatile attractants (*Bjostad and Hibbard, 1992*). Methyl anthranilate, on the other hand, has been shown to repel WCR larvae (*Bernklau et al., 2016b*; *Bernklau et al., 2019*). Many other leaf- and root-feeding herbivores are known to respond to plant volatiles other than $CO_2$ (*Bruce et al., 2005*). Given the low reliability of $CO_2$ as a host-specific cue, it is probably not surprising that WCR, as a highly specialized maize feeder, can use other volatile cues to locate host plants. Integrating

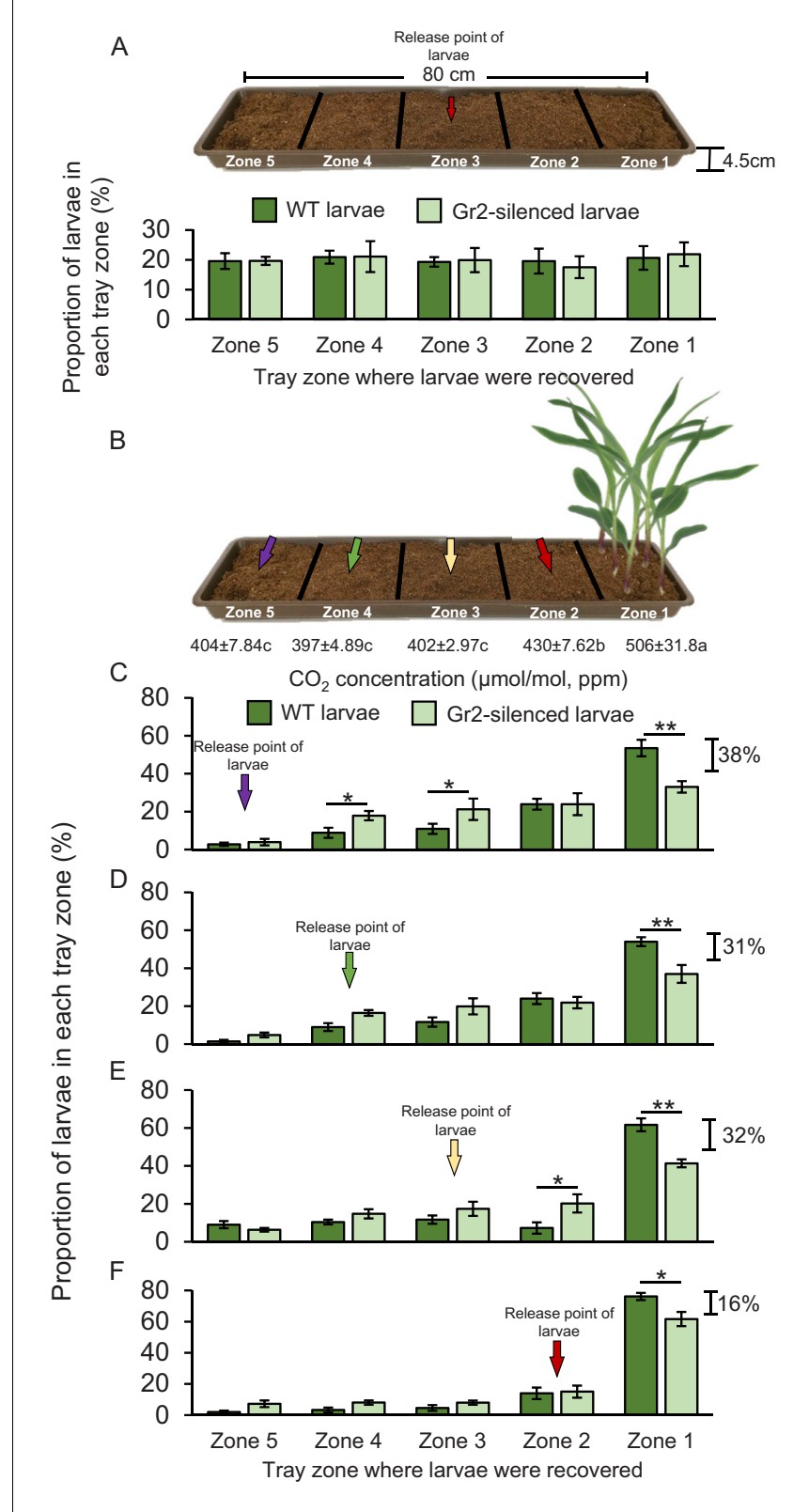

**Figure 7.** Root-associated $CO_2$ is used by western corn rootworm (WCR) larvae for host location in a distance-specific manner. (A) Mean (± SEM) proportion of wild type (WT) (dark green) or *DvvGr2*-silenced (light green) WCR larvae observed in the different tray zones 8 hr after releasing the larvae in the centre of soil-filled trays without plants. Three trays per larval type with 20 larvae each were assayed (n = 3). (B) Schematic representation

*Figure 7 continued on next page*

*Figure 7 continued*

(photomontage) of experimental set-up used to test distance-specific host location abilities of WCR larvae depicting mean (± SEM) $CO_2$ levels detected in the soil gas phase of each tray zone (n = 3–4). Different letters indicate significant differences in $CO_2$ levels (p < 0.05 by one-way ANOVA with Holm's multiple-comparisons test). For detailed data on $CO_2$ levels, refer to *Figure 7—figure supplement 1*. (**C–F**) Mean (± SEM) proportion of WT (dark green) or *DvvGr2*-silenced (light green) WCR larvae observed in the different tray zones 8 hr after releasing the larvae at distances of 64 cm (**C**), 48 cm (**D**), 32 cm (**E**), or 16 cm (**F**) from the plants. Six trays per larval type and distance combination with 20 larvae each were assayed (n = 6). Asterisks indicate statistically significant differences in the proportion of WT and *DvvGr2*-silenced larvae found in each tray zone (*p < 0.05; **p < 0.01 by generalized linear model followed by FDR-corrected post hoc tests). For details regarding the statistical results, refer to *Supplementary file 1*. Raw data are available in *Figure 7—source data 1*.

The online version of this article includes the following source data and figure supplement(s) for figure 7:

**Source data 1.** Raw data for *Figure 7*.
**Figure supplement 1.** Carbon dioxide levels at different sampling points.
**Figure supplement 2.** Root-associated $CO_2$ is required for host location by western corn rootworm (WCR) larvae at long distances.

other volatile cues likely allows WCR larvae to locate maize plants even in the absence of reliable $CO_2$ gradients in the soil, thus increasing the robustness of its foraging behaviour at short distances. An intriguing result in this context is the fact that WCR larvae show the same efficiency in locating maize roots at short distances in the absence of a $CO_2$ gradient, suggesting that this volatile may not play a role as a cue at close range.

Although intact $CO_2$ perception was not required for host location at short distances, it had a strong impact on the capacity of WCR larvae to reach the maize rhizosphere at long distances. A gradient of plant-associated $CO_2$ was detected at distances of up to 32 cm from the plant. When WCR larvae were released at distances greater than 32 cm, they still managed to locate plants in a *DvvGr2*-dependent manner. This result can be explained by random movement, where the larvae move randomly until they encounter a $CO_2$ gradient, or by localized $CO_2$ gradients along preferential gas-phase pathways that may extend beyond 32 cm, or a combination of both. The advantages of $CO_2$ as a host location cue are that it is abundantly produced through respiration by most organisms, is relatively stable (*Jones and Coaker, 1977*; *Li et al., 2016*), and diffuses rapidly in air, water, and soil (*Hashimoto and Suzuki, 2002*; *Ma et al., 2013*). $CO_2$ may thus be a suitable long-range cue to locate organisms with high respiratory rates, such as mammals and heterotroph plant parts, including roots and their associated microbial communities (*Johnson and Nielsen, 2012*). Aboveground insects can be attracted to $CO_2$ traps located as far away as 10 m, and it is estimated that this distance could even be as long as 60 m under optimal environmental circumstances (*Guerenstein and Hildebrand, 2008*; *Zollner et al., 2004*). For belowground insects, this distance is hypothesized to be within the lower centimetre range as $CO_2$ diffusion is substantially decreased within the soil matrix compared to $CO_2$ diffusion in air (*Bernklau et al., 2005*; *Doane et al., 1975*; *Doane and Klingler, 1978*; *Klingler, 1966*). Other volatiles that are less abundant and diffuse even less well through the soil such as (*E*)-β-caryophyllene are unlikely to be detectable at distances of more than 10 cm (*Chiriboga M. et al., 2017*; *Hiltpold and Turlings, 2008*). These volatiles are thus likely useful host location cues at short, but not long, distances in the soil. The finding that WCR integrates $CO_2$ perception with other environmental cues and that attraction to $CO_2$ is context dependent is in line with patterns reported for other insects such as mosquitoes, whose response to stimuli such as colour, temperature, and human body odours is enhanced by $CO_2$ (*McMeniman et al., 2014*; *van Breugel et al., 2015*), and pollinating hawkmoths, which use $CO_2$ as a redundant volatile distance stimulus in a sex-specific manner (*Goyret et al., 2008*).

A recent study shows that a $CO_2$ receptor in *Drosophila* flies is also involved in the detection and behavioural responses to other volatiles (*MacWilliam et al., 2018*). We observed that *DvvGr2*-silenced larvae were repelled by methyl anthranilate, a potent maize root repellent, to a similar extent as WT larvae, suggesting that their sensitivity to this plant volatile is unchanged (*Bernklau et al., 2016b*). In *Drosophila* flies, the $CO_2$ receptor *Gr63a* is required for spermidine attractiveness over short time spans, that is, less than 1 min, but not over longer time spans (hours), when other receptors likely become more important (*MacWilliam et al., 2018*). In the present

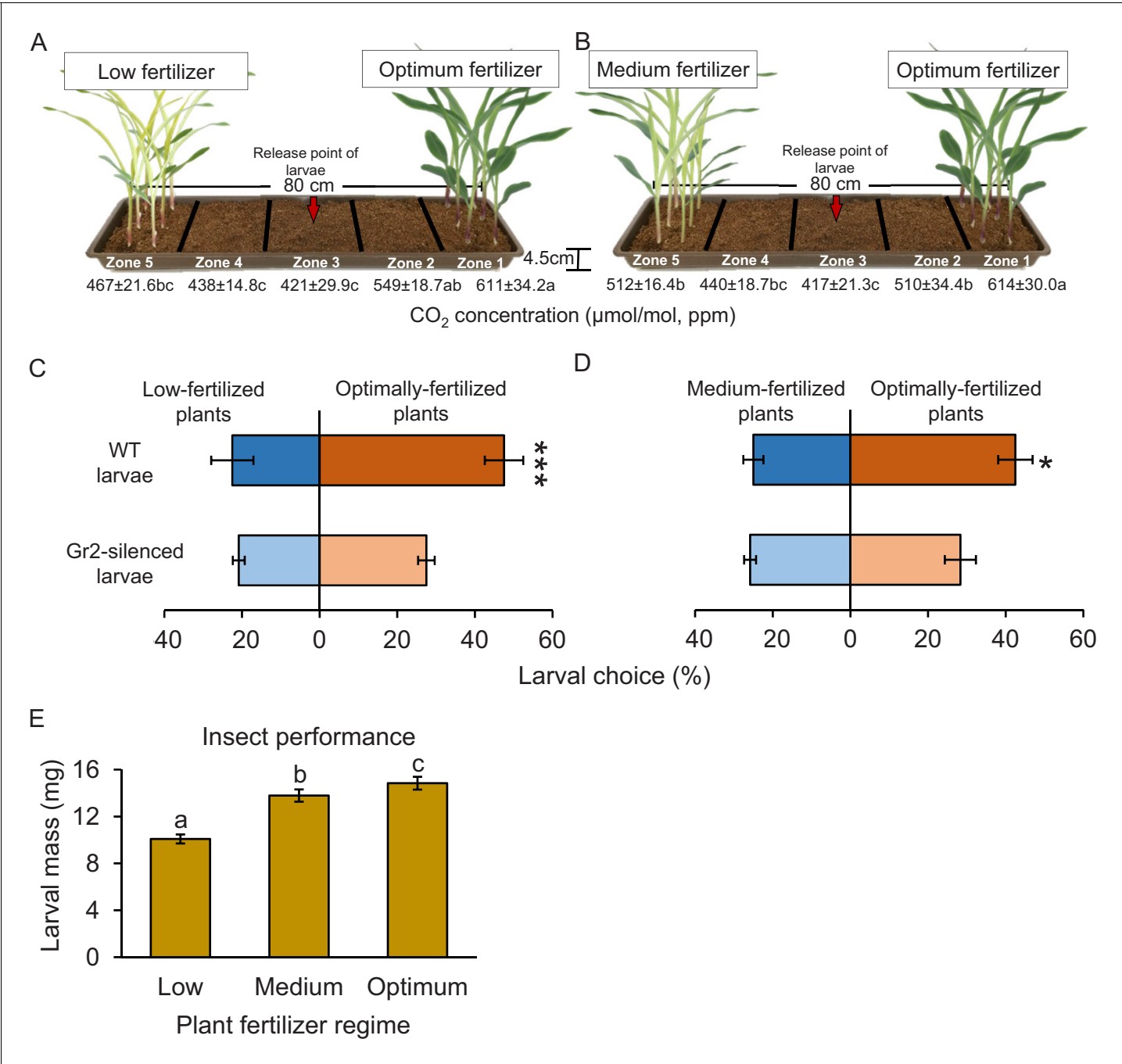

**Figure 8.** $CO_2$ perception increases the location of more suitable host plants. (A, B) Schematic representation (photomontage) of soil-filled trays used to evaluate location of differentially fertilized plants by western corn rootworm (WCR) larvae depicting mean (± SEM) $CO_2$ levels detected in the soil gas phase of each tray zone (n = 10). Different letters indicate statistically significant differences in $CO_2$ levels (p < 0.05 by one-way ANOVA with Holm's multiple-comparisons test). For details regarding $CO_2$ levels, refer to *Figure 8—figure supplement 1*. Mean (± SEM) proportion of WCR larvae recovered close to plants that received low (zone 5) or optimum (zone 1) fertilizer doses (C), or that were recovered close to plants that received medium (zone 5) or optimum (zone 1) fertilizer doses (D) 8 hr after releasing the larvae. Six trays with 20 larvae per tray were assayed (n = 6). Different letters indicate statistically significant differences in larval preferences (*p < 0.05; ***p < 0.001 by generalized linear model followed by FDR-corrected post hoc tests). (E) Mean (± SEM) weight of WCR larvae after 7 days feeding on plants fertilized with low, medium, or optimum fertilizer doses. Twenty solo cups with 4–7 larvae each were assayed (n = 20). Different letters indicate statistically significant differences in larval mass (p < 0.05 by one-way ANOVA followed by Holm's multiple-comparisons tests). For details regarding the statistical results, refer to *Supplementary file 1*. Raw data are available in *Figure 8—source data 1*.

The online version of this article includes the following source data and figure supplement(s) for figure 8:

*Figure 8 continued on next page*

*Figure 8 continued*

**Source data 1.** Raw data for *Figure 8*.
**Figure supplement 1.** Carbon dioxide levels at different sampling points.

experiments, WCR behaviour was evaluated after one or more hours. The $CO_2$ scrubber experiment provides further evidence that the foraging patterns observed in this study are not due to different sensitivity of *DvvGr2*-silenced larvae to other root volatiles.

Apart from acting as a long-distance host location cue, $CO_2$ also links plant fertilization to herbivore behaviour by guiding WCR to well-fertilized plants. As WCR larvae are resistant to root defences of maize (*Robert et al., 2012b*), it is likely to benefit from increased fertilization, independently of the plant's defensive status. As the plant nutritional status and host quality for WCR larvae are associated with higher $CO_2$ release from the roots, following the highest concentrations of $CO_2$ in the soil may be adaptive for the herbivore as it may increase its chance not only to find a maize plant per se but also to identify a plant that has the resources to grow vigorously and that is a better host. More experiments are needed to confirm this hypothesis as in the current set-up the larvae may have followed the only available $CO_2$ gradient close to their release point rather than having made a choice between two gradients. However, given the dose-dependent responses of WCR, preferential orientation towards plants surrounded by higher $CO_2$ levels appears likely. Well-fertilized maize plants increase photosynthesis and biomass production, which results in higher $CO_2$ release from the roots (*Zhu and Lynch, 2004*). WCR larvae are specialized maize pests that have evolved with intense maize cultivation in the corn belt of the US (*Gray et al., 2009*) and are resistant to maize defence metabolites (*Robert et al., 2012b*). Following the strongest $CO_2$ gradient in an equally spaced maize monoculture may indeed be a useful strategy for this root feeder to locate suitable food sources. An association between $CO_2$ emission and food-source profitability was also suggested for *Datura* flowers, which emit the highest level of $CO_2$ in times when nectar is most abundant (*Guerenstein et al., 2004*; *Thom et al., 2004*). These findings support the general hypothesis that $CO_2$ is a marker of metabolic activity that allows for an assessment of the vigour and profitability of a wide variety of hosts. The impact of $CO_2$ for the distribution of root herbivores such as the WCR in heterogeneous environments remains to be determined. Based on our results, we expect plant-associated $CO_2$ to contribute to uneven herbivore distribution and to aggregation on plants with a good nutritional status within monocultures.

In summary, this work demonstrates how a herbivore uses its capacity to perceive $CO_2$ to locate host plants. Volatiles other than $CO_2$ are also integrated into host-finding behaviour in the soil, but their effects are more important at short than at long distances. Random movement in the soil may help this root herbivore to increase its capacity to find host cues at even greater distances. Thus, evidence is now accumulating that $CO_2$ acts as an important host location cue in different insects, likely because of its unique role as a highly conserved long-range marker of metabolic activity within complex sensory landscapes.

## Materials and methods

### Plants and planting conditions

Maize seeds (*Zea mays* L., var. Akku) were provided by Delley Semences et Plantes SA (Delley, Switzerland). Seedlings were grown under greenhouse conditions (23 ± 2°C, 60% relative humidity, 16:8 h L/D, and 250 mmol/m$^2$/s[1] additional light supplied by sodium lamps). Plantaaktiv 16+6+26 Typ K fertilizer (Hauert HBG Dünger AG, Grossaffoltern, Switzerland) was added twice a week after plant emergence following the manufacturer's recommendations. The composition of the fertilizer is: total nitrogen (N) 16%, nitrate 11%, ammonium 5%, phosphate ($P_2O_5$) 6%, potassium oxide ($K_2O$) 26%, magnesium oxide (MgO) 3.3%, boron (B) 0.02%, copper (Cu, EDTA-chelated) 0.04%, iron (Fe, EDTA-chelated) 0.1%, manganese (Mn, EDTA-chelated) 0.05%, molybdenum (Mo) 0.01%, and zinc (Zn, EDTA-chelated) 0.01%. When plants were used as insect food, seedlings were germinated in vermiculite (particle size: 2–4 mm; tabaksamen, Switzerland) and used within 4 days after germination.

## Insects and insect rearing

*Diabrotica virgifera virgifera* (WCR) insects used in this study were derived from a non-diapausing colony reared at the University of Neuchâtel. The eggs used to establish the colony were supplied by USDA-ARS-NCARL, Brookings, SD. New insects of the same origin are introduced into the colony every 3–6 months. Upon hatching, insects were maintained in organic soil (Selmaterra, Bigler Samen AG, Thun, Switzerland) and fed freshly germinated maize seedlings (var. Akku).

## Identification of $CO_2$ receptor genes

To identify $CO_2$ receptor orthologues in WCR, we used $CO_2$ receptor-encoding gene sequences of *T. castaneum* and several sequences from other insects as queries against publicly available WCR genome sequences (NCBI accession: PXJM00000000.2) using the National Center for Biotechnology Information Basic Local Alignment Search Tool (NCBI BLAST) (*Robertson and Kent, 2009*; *Wang et al., 2013*; *Xu and Anderson, 2015*). The full gene sequences can be retrieved from the NCBI databank using the following accession numbers: XM_028276483.1 (*DvvGr1*), XM_028280521.1 (*DvvGr2*), and XM_028272033.1 (*DvvGr3*). These gene sequences were translated to obtain protein sequences. The obtained protein sequences and the protein sequences of $CO_2$ receptors from different insects were used to infer evolutionary relationships using the neighbor-joining method in MEGA 7 (*Kumar et al., 2016*; *Robertson and Kent, 2009*; *Rodrigues et al., 2016*; *Saitou and Nei, 1987*). The optimal tree with the sum of branch length = 4.44068889 is provided in *Figure 2A*. The percentage of replicate trees in which the associated taxa clustered together in the bootstrap test (100 replicates) are shown next to the branches (*Felsenstein, 1985*). The tree is drawn to scale, with branch lengths in the same units as those of the evolutionary distances used to infer the phylogenetic tree. The evolutionary distances were computed using the Poisson correction method (*Zuckerkandl and Pauling, 1965*) and are in the units of the number of amino acid substitutions per site. A total of 242 amino acid positions were included in the final data set. Graphical representation and edition of the phylogenetic tree were performed with the Interactive Tree of Life (version 3.5.1) (*Letunic and Bork, 2016*). Protein tertiary structures and topologies were predicted using Phyre2 (*Kelley et al., 2015*).

## Production of dsRNA

*Escherichia coli* HT115 were transformed with recombinant L4440 plasmids that contained a 211–240 bp long gene fragment targeting one of the three $CO_2$ receptors. Cloned nucleotide sequences were synthetized de novo (Eurofins, Germany). To induce the production of dsRNA, an overnight bacterial culture was used to inoculate fresh Luria–Berthani broth (25 g/L, Luria/Miller, Carl Roth GmbH, Karlsruhe, Germany). Once the bacterial culture reached an $OD_{600}$ of 0.6–0.8, it was supplemented with isopropyl β-D-1-thiogalactopyranoside (Sigma-Aldrich, Switzerland) at a final concentration of 2 mM. Bacterial cultures were incubated at 37°C in an orbital shaker (Ecotron, Infors HT, Bottmingen, Switzerland) at 130 rpm for 16 additional hours. Bacteria were harvested by centrifugation (2000 rpm, 10 min) using a top bench centrifuge (IEC Centra GP6R, Thermo Fisher Scientific, Waltham, MA, USA) and stored at −20°C in a freezer (Bosch, Gerlingen, Germany) for further use (*Kim et al., 2015*).

## Gene silencing experiments

To induce gene silencing in WCR, 6–10 second instar WCR larvae were released in solo cups (30 ml, Frontier Scientific Services, Inc, Germany) containing approximately 2 g of autoclaved soil (Selmaterra, Bigler Samen AG, Thun, Switzerland) and 2–3 freshly germinated maize seedlings. Maize seedlings were coated with 1 ml of bacterial solution containing approximately 200–500 ng of dsRNA targeting the different $CO_2$ receptor genes. As controls, larvae were fed with bacteria-producing dsRNA-targeting GFP genes, which are absent in the WCR genome (*Rodrigues et al., 2016*). dsRNA was produced as described above. Fresh bacteria and seedlings were added to solo cups every other day for three consecutive times. Two days after the last dsRNA/bacteria application, larvae were collected and used for experiments.

## Gene expression measurements

Total RNA was isolated from approximately 10 mg of frozen, ground, and homogenized WCR larval tissue (3–7 larvae per biological replicate, n = 8) using the GenElute Universal Total RNA Purification Kit (Sigma-Aldrich, St. Louis, MO, USA). A NanoDrop spectrophotometer (ND-1000, Thermo Fisher Scientific, Waltham, MA, USA) was used to estimate RNA purity and quantity. DNase-treated RNA was used as template for reverse transcription and first-strand cDNA synthesis with PrimeScript Reverse Transcriptase (Takara Bio Inc, Kusatsu, Japan). DNase treatment was carried out using the gDNA Eraser (Perfect Real Time) following manufacturer's instructions (Takara Bio Inc). For gene expression analysis, 2 µl of undiluted cDNA (i.e., the equivalent of 100 ng total RNA) served as template in a 20 µl qRT-PCR using the TB Green Premix Ex Taq II (Tli RNaseH Plus) kit (Takara Bio Inc) and the Roche LightCycler 96 system (Roche, Basel, Switzerland), according to manufacturer's instructions. Transcript abundances of the following WCR genes were analysed: *DvvGr1, DvvGr2*, and *DvvGr3* (*Rodrigues et al., 2016*). *Actin* was used as reference gene to normalize expression data across samples. Relative gene expression levels were calculated by the $2^{-\Delta\Delta Ct}$ method (*Livak and Schmittgen, 2001*). The following primers were used: DvvGr1-F CGTTAATTTAGCTGCTGTGG, DvvGr1-R GTTTTCTGTTGCTAGAGTTGC, DvvGr2-F GAACTAAGCGAGCTCCTCCA, DvvGr2-R CAGAAGCACCATGCAATACG, DvvGr3-F GCAACGCTTTCAGCTTTACC, DvvGr3-R GTGCATCGTCATTCATCCAG, DvvActin-F TCCAGGCTGTACTCTCCTTG, and DvvActin-R CAAGTCCAAACGAAGGATTG.

## $CO_2$ measurements

$CO_2$ was quantified by an infrared $CO_2$ gas analyser or by gas chromatography coupled to a flame ionization detector (GC-FID). In the first case, air samples were collected by a micropump (Intelligent Subsampler TR-SS3, Sable Systems International, Las Vegas, NV, USA) connected to an airstream selector (RM8 Intelligent Multiplexer, V5, Sable Systems International, Las Vegas, NV, USA) controlled by a computer via a Universal Interface (UI-2) and the Expedata software version 1.2.6 (Sable Systems International, Las Vegas, NV, USA). The sampled air passed through an infrared $CO_2$ gas analyser (LI 7000, Li-Cor Inc, Lincoln, NE, USA) (*Frei et al., 2017*). In the second case, air samples were collected by using a syringe equipped with a Luer connector, a stopcock valve, and a needle. Also, 3 ml of air samples were immediately injected and $CO_2$ was analysed by a GC-FID (Shimadzu GC-8) equipped with a methanizer (VWR, Radnor, PA, USA) and a Poropack N column. Nitrogen was used as carrier gas (400 kPa). After injection, the column temperature was maintained at 100°C for 1.3 min, and then increased to 130°C at a rate of 30°C/min and maintained at this temperature for 4 min. The FID temperature was set at 250°C.

## Root volatile measurements

To measure root volatiles, three 4-day-old maize seedling were transplanted into moist white sand (Migros, Switzerland) in spherical glass pots (7 cm diameter, Verre and Quartz Technique SA, Neuchâtel, Switzerland). To boost volatile release, the roots were damaged mechanically before transplantation by briefly twisting them. The pots were wrapped in aluminium foil. Clean humidified air was pushed through the pots at a rate of 1 l·min$^{-1}$ and pulled through Porapak filters (25 mg of Porapak adsorbent, 80–100 mesh; Alltech Assoc., Deerfield, IL, USA) at a rate of 0.6 L·min$^{-1}$. Root volatiles were collected over 6 hr. After this period, the filters were eluted with 150 µl of dichloromethane, and N-octane and nonyl-acetate (Sigma, Buchs, Switzerland) were further added as internal standards (200 ng in 10 µl dichloromethane). The root volatiles were analysed by gas chromatography coupled to mass spectrometry (Agilent 7820A GC coupled to an Agilent 5977E MS, Agilent Technologies, Santa Clara, CA, USA). The aliquot was injected in the injector port (230°C) and pulsed in a spitless mode onto an apolar column (HP-5MS 5% Phenyl Methyl Silox, 30 m × 250 µm internal diameter × 0.25 µm film thickness, J&W Scientific, Agilent Technologies SA, Basel, Switzerland). Helium at a constant flow of 1 ml·min$^{-1}$ (constant pressure 8.2317 psi) was used as carrier gas. After injection, the column temperature was maintained at 40°C for 3.5 min, and then increased to 100°C at a rate of 8°C/min and subsequently at 5°C/min to 230°C, followed by a post run of 3 min at 250°C. Volatile identification was obtained by comparing mass spectra with those of the NIST17 Mass Spectra Library, and relative quantities for the major compounds were calculated based on the peak areas of the internal standards.

## Belowground olfactometer experiments with *DvvGr1-*, *DvvGr2-*, and *DvvGr3*-silenced WCR larvae

To determine whether silencing putative $CO_2$ receptor genes impairs the ability of WCR larvae to behaviourally respond to $CO_2$, we silenced *DvvGr1*, *DvvGr2*, and *DvvGr3* as described above and evaluated larvae responses to $CO_2$ in dual-choice experiments using belowground olfactometers. Larvae that were fed bacteria that express dsRNA that targets GFP were used as controls (herein referred to as wild type larvae; WT). The belowground olfactometers consist of two L-shaped glass pots (5 cm diameter, 11 cm deep) connected to a detachable central glass tube (24–29 mm diameter, 8 cm in length) by detachable Teflon connectors (Verre and Quartz Technique SA, Neuchâtel, Switzerland) (*Figure 2B*). The Teflon connectors contained wire mesh screens (2300 mesh, Small Parts Inc, Miami Lakes, FL, USA) to restrain the larvae from moving into the plants. The central glass tubes remained empty to only allow volatile compounds to diffuse through the central glass tubes. The central glass tubes have an access port in the middle to allow the release of insects (*Figure 2B*). Insects can freely move inside the central glass tube (8 cm in length) and reach the metal wire screens. For further technical specifications regarding the belowground olfactometers, refer to *Rasmann et al., 2005* and *Robert et al., 2012a*. To increase $CO_2$ levels in one side of the olfactometers, we used carbonated water as a $CO_2$ source (*Bernklau and Bjostad, 1998a*; *Huang et al., 2017*; *Jewett and Bjostad, 1996*). For this, a plastic cup containing 50 ml of carbonated water (Valais, Aproz Sources Minérales, Aproz, Switzerland) was placed in one L-shaped glass pot and a plastic cup containing 50 ml of distilled water was placed in the opposite pot. L-shaped glass pots did not contain any substrate to allow $CO_2$ to freely diffuse into the central glass tubes (*Figure 2B*). Ten minutes after placing the cups into the L-shaped glass pots, L-shaped glass pots were connected to the central glass tubes and six larvae were released in the middle of the olfactometer (red arrow, *Figure 2B*). Larval positions were recorded 1 hr after their release. Insect preference for a given treatment was considered when the larvae were found at a distance of 1 cm or less from the odour source; this is 1 cm apart from the wire mesh. Seven olfactometers per larval type and experiment were assessed. Olfactometers were covered with aluminium foil to reduce light disturbance to the larvae. Aluminium foil was removed shortly before evaluating larval positions. The experiment was repeated twice. $CO_2$ levels on each arm of the olfactometer were measured by GC-FID as described above. Air samples to determine $CO_2$ concentrations were collected by using a syringe equipped with a Luer connector, stopcock valve, and needle. Prior to sampling, we first connected the Teflon connectors to the L-shaped glass pots and closeed them with parafilm. Then, we placed a plastic cup containing either 50 ml of carbonated water or 50 ml of distilled water. Ten minutes after, we pierced the parafilm with the needle and collected 3 ml of air samples using the syringe and immediately injected the samples to the GC-FID for $CO_2$ measurements.

## Insect responses to methyl anthranilate

To determine whether silencing the *DvvGr2* carbon dioxide receptor affects larval responses to a plant volatile other than $CO_2$, we evaluated larval responses to methyl anthranilate. For this, we evaluated insect preferences for seedling roots or for seedling roots placed next to a filter paper disc treated with methyl anthranilate following a similar experimental procedure as described by *Bernklau et al., 2019*. To this end, either five 2nd–3rd instar WT WCR larvae or five 2nd–3rd instar *DvvGr2*-silenced WCR larvae were released in the middle of a moist sand-filled Petri plate (9 cm diameter, Greiner Bio-One GmbH, Frickenhausen, DE) where they encountered two 3-day-old maize seedling roots in one side or two 3-day-old maize seedlings roots and a filter paper disc (0.5 cm diameter, Whatman no. 1, GE Healthcare Life Sciences, UK) treated with 10 µl of methyl anthranilate solution (10 mg/ml of water) in the opposite side. Methyl anthranilate was purchased from Sigma (CAS: 134-20-3; Sigma Aldrich Chemie, Switzerland). Sand layers were 3–4 mm high and allowed the larvae to move freely in and on the substrate. Ten Petri plates per larval type with five larvae each were evaluated (n = 10). Petri plates were covered with black plastic sheets to avoid light disturbance to the insects. Larval positions were recorded 1 hr after releasing the larvae. Insect preference for a given treatment was considered when the larvae were found on the roots or in contact with the filter paper discs.

### Insect responses to Fe(III)(DIMBOA)$_3$

To determine whether silencing the *DvvGr2* carbon dioxide receptor affects larval responses to plant metabolites other than $CO_2$, we evaluated larval responses to Fe(III)(DIMBOA)$_3$. For this, we evaluated insect preferences for filter paper discs impregnated with Fe(III)(DIMBOA)$_3$ or for filter paper discs treated with water following the procedure described by *Hu et al., 2018* with minor modifications. To this end, we released either six 2nd–3rd instar WT WCR larvae or six 2nd–3rd instar *DvvGr2*-silenced WCR larvae in the middle of a moist sand-filled Petri plate (6 cm diameter, Greiner Bio-One GmbH, Frickenhausen, DE) where they encountered a filter paper disc (0.5 cm diameter, Whatman no. 1, GE Healthcare Life Sciences, UK) treated with 10 μl of Fe(III)(DIMBOA)$_3$ (1 μg/ml of water) or, on the opposite side, a filter paper disc treated with water only. Twenty Petri plates with six larvae each were evaluated (n = 20). Fe(III)(DIMBOA)$_3$ was prepared fresh by mixing FeCl$_3$ and DIMBOA at a 1:2 ratio as described by *Hu et al., 2018*. Petri plates were covered with black plastic sheets to avoid light disturbance to the insects. Larval preferences were recorded 1 hr after releasing the larvae. Insect preference for a given treatment was considered when the larvae were found in contact with the filter paper discs.

### Insect responses to soluble sugars

To determine whether silencing the *DvvGr2* carbon dioxide receptor affects larval responses to plant metabolites other than $CO_2$, we evaluated larval responses to soluble sugars. For this, we evaluated insect preferences for a mixture of glucose, fructose, and sucrose following the procedure described by *Bernklau et al., 2018* with minor modifications. Briefly, we released either six 2nd–3 instar WT WCR larvae or six 2nd–3rd instar *DvvGr2*-silenced WCR larvae in the middle of a moist sand-filled Petri plate (6 cm diameter, Greiner Bio-One GmbH, Frickenhausen, DE) where they encountered a filter paper disc (0.5 cm diameter, Whatman no. 1, GE Healthcare Life Sciences, UK) treated with 10 μl of a mixture of glucose, fructose, and sucrose (30 mg/ml of each sugar), or, on the opposite side, a filter paper treated with water only. Twenty Petri plates with six larvae each were assayed (n = 20). Petri plates were covered with black plastic sheets to avoid light disturbance to the insects. Larval preferences were recorded 3 hr after release. Insect preference for a given treatment was considered when the larvae were found in contact with the filter paper discs.

### Dose-dependent insect responses to $CO_2$

To determine whether silencing the *DvvGr2* carbon dioxide receptor affects larval responses to $CO_2$ and to determine the range of behaviourally active $CO_2$ concentrations, we evaluated larval responses to different concentrations of $CO_2$ in dual-choice experiments using belowground olfactometers. $CO_2$ levels were increased in one side of the olfactometer by delivering $CO_2$-enriched synthetic air (1% $CO_2$, Carbagas, Switzerland). For this, the L-shaped glass pots were closed on top using parafilm during $CO_2$ delivery. A manometer connected to the synthetic air bottle allowed to fine-tune $CO_2$ delivery rates and concentrations. $CO_2$ levels were measured using a gas analyser (Li7000, Li-Cor Inc, Lincoln, NE, USA) as described above. $CO_2$ levels were increased at different levels from 22 to 1832 ppm above ambient $CO_2$ levels. Once the desired $CO_2$ concentrations were reached, $CO_2$ delivery was terminated, and the olfactometers were assembled by connecting two L-shaped glass pots to a central glass tube as described above. Immediately after this, either six 2nd–3rd instar WT WCR larvae or six 2nd–3rd instar *DvvGr2*-silenced WCR larvae were released in the middle of the central glass tubes. Larval positions were evaluated within 10 min of release. Experiments were conducted in a dark room to reduce light disturbance to the larvae. Red-light headlamps were used by the experimenters during the experiment. Insect preference for a given treatment was considered when the larvae were found at a distance of 1 cm or less from the odour source; this is 1 cm apart from the wire mesh. Three olfactometers per larval type and six larvae per olfactometer were assayed (n = 3).

### Insect motility and speed experiments

To determine whether silencing the *DvvGr2* carbon dioxide receptor affects larval motility and speed, larval behaviour and the trajectories followed by individual larvae in open Petri plates that contained either maize root pieces or that were outfitted with a $CO_2$ point releaser in the middle were evaluated. In the first experiment, two root pieces (3–4 cm long) of 4-day-old maize seedlings

were placed at the rim of a Petri plate lined with moist filter paper. Then, on the opposite rim (i.e., 9 cm apart), either one 2nd–3rd instar WT WCR larvae or one 2nd–3rd instar *DvvGr2*-silenced WCR larvae was released. The trajectories followed by the insects and the time required to reach the roots were evaluated. The trajectories followed by the insects were drawn on circular pieces of papers of 9 cm diameter. Six Petri plates with one larva each were assayed (n = 6). In the second experiment, $CO_2$ point releasers were installed in the centre of Petri plates (9 cm diameter, Greiner Bio-One, Austria). For this, Petri plates were pierced with a hot metal needle. Then, another needle that released $CO_2$ at 581 ppm, resulting in $CO_2$ concentrations 60 ppm above ambient $CO_2$ levels, was inserted in the resulting whole. $CO_2$ levels were adjusted using a manometer connected to the synthetic air bottle. $CO_2$ levels were measured using a gas analyser (Li7000, Li-Cor Inc, Lincoln, NE, USA) as described above. Once the desired $CO_2$ concentration was reached, either one 2nd–3rd instar WT WCR larva or one 2nd–3rd instar *DvvGr2*-silenced WCR larva was released at the rim of the Petri plates (i.e., 4.5 cm apart from the $CO_2$ point releaser). Petri plates were lined with moist filter paper (9 cm diameter, GE Healthcare, UK). Insect behaviour was observed for 3 min, the trajectories followed by the insects during this time were drawn on circular pieces of papers of 9 cm diameter, and the time spent in close contact with the $CO_2$ point releaser was quantified. Six Petri plates with one larva each were assayed (n = 6). Both experiments were conducted in a dark room to reduce light disturbance to the larvae. Red-light headlamps were used by the experimenters during the experiment. The pieces of paper with the drawings of the insect trajectories were scanned. The resulting images were analysed in ImageJ 1.53a to determine the distances crawled by the insects.

## Host location experiments using belowground olfactometers

To determine the importance of plant-associated $CO_2$ for host location by WCR larvae and to test for distance-specific effects, we evaluated host location ability of WCR larvae in dual-choice experiments using belowground olfactometers (*Figure 5*). To specifically investigate the importance of plant-associated $CO_2$ for host location, preferences of $CO_2$-sensitive and $CO_2$-insensitive insects for intact plant odours and for plant odours without $CO_2$ were evaluated. $CO_2$ was experimentally removed using soda lime (Carl Roth, Karlsruhe, Germany). For this, layers of 5 g of soda lime granules (2–4 mm) were placed between the metal wire screens of the Teflon connectors and the sand contained in the L-shaped glass pots. To test for distance-specific effects, we used two olfactometer types that were otherwise the same but differed in the length of their arms (*Figure 5*). General specifications of the olfactometers are described above. One set of olfactometers has short arms that allowed for the release of larvae at 9 cm (*Figure 5A*) from plant volatile sources, and the other one has long arms that allow to release the larvae at 18 cm (*Figure 5B*) from plant volatiles sources. Insect larvae can move freely into the central glass tube and reach the metal wire screens located at both ends of the central glass tube. L-shaped glass pots were covered with aluminium foil, filled with sand, and one 3-week-old maize plant was transplanted 48 hr before the experiments. L-shaped glass pots without plants were treated similarly. Olfactometers remained detached until shortly before the experiments. Thirty minutes before the experiments, soda lime layers were applied and one L-shaped glass pot with a plant and one L-shaped glass pot without a plant were connected to the central glass tubes. Soda lime layers were used in both sides of the olfactometers. Olfactometers without soda lime served as controls. Then, either six 2nd–3rd instar WT WCR larvae or six 2nd–3rd instar *DvvGr2*-silenced WCR larvae were released. Olfactometers were covered with aluminium foil to reduce light disturbance to the larvae. Aluminium foil was removed shortly before evaluating larval positions. Larval positions were recorded 1 hr after their release. Insect preference for a given treatment was considered when the larvae were found at least 1 cm from the odour source; this is 1 cm apart from the wire mesh. Four olfactometers with six larvae each were assayed (n = 4). $CO_2$ levels were measured using a gas analyser (Li7000, Li-Cor Inc, Lincoln, NE, USA) as described above.

## Host location experiments using soil-filled trays

To determine the importance of plant-associated $CO_2$ for host location by WCR larvae and to test for distance-specific effects, host location by $CO_2$-sensitive and $CO_2$-insensitive insects was evaluated in soil-filled trays (*Figure 7*). For this, four or five 2-to 3-week-old maize plants were transplanted into custom-made fabric pockets (12 × 3 × 5 cm) made out of volatile-permeable fabrics

(Trenn-Vlies, GeoTex Windhager, Switzerland) filled with soil (Selmaterra, Bigler Samen AG, Thun, Switzerland). The plants were transplanted into the fabric pockets, and the fabric pockets were placed in a corner of plastic trays (80 cm × 15 cm × 4.5 cm) (Migros Do it + Garden, Switzerland) containing soil 24 hr before the experiments. Then, 20 second instar WT WCR larvae or 20 second instar *DvvGr2*-silenced WCR larvae were released at 16, 32, 48, or 64 cm from the plants. Larval release points are indicated by arrows (*Figure 7B*). Eight hours after releasing the larvae, their positions were recorded by carefully removing the soil at each tray zone and inspecting it in search for the insects. Six trays per larval type and distance with 20 larvae each were assayed (n = 6). $CO_2$ levels in the soil gas phase of each tray zone were measured by GC-FID as described above.

### Host location experiments with differentially fertilized plants

Optimally fertilized plants emit higher levels of respiratory $CO_2$ than suboptimally fertilized plants (*Zhu and Lynch, 2004*). To test whether WCR larvae orient towards and prefer optimally fertilized maize plants and to evaluate the importance of $CO_2$ in this context, host location by $CO_2$-sensitive and $CO_2$-insensitive insects and preference for plants that were differentially fertilized was evaluated. To this end, maize plants were fertilized with three doses of fertilizer: 0.1% (optimally fertilized), 0.05% (medium fertilized), or 0.01% (low fertilized). For this, Plantaaktiv 16+6+26 Typ K fertilizer (Hauert HBG Dünger AG, Grossaffoltern, Switzerland) was dissolved to the abovementioned concentrations and applied following manufacturer's indications. Fertilizer macro- and micronutrient composition are described above (see 'Plants and planting conditions'). For the choice experiments, plants were grown under the different fertilizer regimes. Twenty-four hours before the choice experiment, five 15-day-old, optimally fertilized plants were re-planted into fabric pockets (described above) and transferred to one corner of an 80 cm long plastic tray (Migros Do it + Garden, Switzerland) filled with soil (*Figure 8A, B*). At the opposite side, plants grown either under low or medium fertilizer regimes were equally re-planted and transferred. Then, 20 second instar WCR larvae were released in the middle of the tray (zone 3, red arrows). Six independent trays per larval type and fertilizer regime pair were evaluated (n = 6). Eight hours after releasing the larvae, larval positions were recorded.

### Effect of plant nutritional status on larval growth

To test whether larval preference for optimally fertilized plants is reflected in their growth, we measured larval weights of larvae feeding on roots of plants that were grown under the different fertilizer regimes. Fertilizer regimes are described above. For this, seven 1st instar larvae were released into solo cups (30 ml, Frontier Scientific Services, Inc, DE) containing approximately 2 g of organic soil (Selmaterra, Bigler Samen AG, Thun, Switzerland). Fresh roots of 15-day-old plants were provided everyday ad libitum. Roots were washed thoroughly to remove fertilizer traces from the root surface. Twenty solo cups per treatment were included (n = 20). Eight days after the beginning of the experiment, larvae were weighed using a micro balance. Four to seven larvae were recovered per experimental unit at the end of the experiment.

### Statistical analyses

Differences in gene expression levels, larval performance, and carbon dioxide concentrations were analysed by either Student's *t* tests or by one-way ANOVA using Sigma Plot 12.0 (SystatSoftware Inc, San Jose, CA, USA). Normality and equality of variance were verified using Shapiro–Wilk, Levene's, and Brown–Forsythe tests. Holm–Sidak post hoc tests were used for multiple comparisons. Data sets from experiments that did not fulfil the assumptions for ANOVA were natural log-, root square-, or rank-transformed before analysis. Carbon dioxide concentrations and differences in larval preference were assessed using GLM under binomial distribution and corrected for overdispersion with quasi-binomial function when necessary followed by analysis of deviance and FDR-corrected post hoc tests. All analyses were followed by residual analysis to verify the suitability of the error distribution and model fitting. All The above analyses were conducted using R 3.2.2 (43) using the packages 'lme4', 'car', 'lsmeans', and 'RVAideMemoire' (*Bates et al., 2014*; *Fox and Weisberg, 2011*; *Herve, 2015*; *Lenth, 2016*; *R Development Core Team, 2014*).

## Acknowledgements

We thank Evangelia Vogiatzaki and Celine Terrettaz for technical assistance, and the members of the Research Section Biotic Interactions of the Institute of Plant Sciences of the University of Bern, Switzerland, for their support and helpful discussions. This project was supported by the University of Bern, a European Union Horizon 2020 *Marie Sklodowska-Curie Action* (MSCA) Individual Fellowship (grant no. 794947 to BCJS), and the Swiss National Science Foundation (grant no. 155781 to ME, grant no. 186094 to RARM).

## Additional information

### Funding

| Funder | Grant reference number | Author |
|---|---|---|
| H2020 Marie Skłodowska-Curie Actions | 794947 | Bernardus CJ Schimmel |
| Schweizerischer Nationalfonds zur Förderung der Wissenschaftlichen Forschung | 155781 | Matthias Erb |
| Schweizerischer Nationalfonds zur Förderung der Wissenschaftlichen Forschung | 186094 | Ricardo AR Machado |

The funders had no role in study design, data collection and interpretation, or the decision to submit the work for publication.

### Author contributions

Carla CM Arce, Resources, Formal analysis, Investigation, Methodology, Writing - original draft, Writing - review and editing; Vanitha Theepan, Geoffrey Jaffuel, Investigation; Bernardus CJ Schimmel, Investigation, Writing - review and editing; Matthias Erb, Conceptualization, Formal analysis, Supervision, Funding acquisition, Methodology, Writing - original draft, Writing - review and editing; Ricardo AR Machado, Conceptualization, Resources, Data curation, Formal analysis, Supervision, Funding acquisition, Validation, Investigation, Visualization, Methodology, Writing - original draft, Writing - review and editing

### Author ORCIDs

Carla CM Arce https://orcid.org/0000-0002-1713-6970
Matthias Erb https://orcid.org/0000-0002-4446-9834
Ricardo AR Machado https://orcid.org/0000-0002-7624-1105

### Decision letter and Author response

Decision letter https://doi.org/10.7554/eLife.65575.sa1
Author response https://doi.org/10.7554/eLife.65575.sa2

## Additional files

### Supplementary files

• Supplementary file 1. Description and results of the statistical analysis methods used to analyse the data of this study.

• Transparent reporting form

### Data availability

All data that support the conclusions are given in form of figures/tables within the manuscript. Raw data sets are provided as source data files.

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
