## [Decision Letter]

**Acceptance summary:**

Your manuscript describes an exciting study on the role of CO_2_ in the host-searching by a soil-borne insect herbivore, especially because of generating insect larvae with reduced expression levels of CO_2_ receptor-encoding genes through RNAi. This study provides very interesting, novel insights into the use of a compound generally produced by living organisms in the process of searching for a host plant. Moreover, the use of different experimental setups at different spatial scales adds to the value of the study.

**Decision letter after peer review:**

[Editors’ note: the authors submitted for reconsideration following the decision after peer review. What follows is the decision letter after the first round of review.]

Thank you for submitting your work entitled "Plant-derived CO_2_ mediates long-distance host location and quality assessment by a root herbivore" for consideration by *eLife*. Your article has been reviewed by three peer reviewers, one of whom is a member of our Board of Reviewing Editors, and the evaluation has been overseen by a Senior Editor. The following individual involved in review of your submission have agreed to reveal their identity: Andre Kessler (Reviewer #3).

Our decision has been reached after consultation between the reviewers. Based on these discussions and the individual reviews below, we regret to inform you that your work will not be considered further for publication in *eLife*.

Although all three reviewers were highly interested in the study and consider the generation of the silenced larvae a very interesting approach to studying the role of CO_2_ in host-plant selection by larvae of the western corn rootworm, most of the experiments addressing their behavior do not unequivocally lead to the conclusions drawn in the manuscript. This relates especially to the role of CO_2_ in long-range attraction, the relative role of CO_2_ and other volatiles, the source of the CO_2_. Resolving these issues would require extensive additional experimentation and that is something *eLife* does not ask from authors. In addition the methods section lacks many important details to allow careful assessment of the data presented. Therefore, the reviewers unanimously conclude that this manuscript is not acceptable for publication in *eLife*.

Reviewer #1:

This manuscript has an interesting topic, presenting a study on the effect of CO_2_ on the behavior of a root-feeding insect, larvae of the western corn rootworm. Despite ample research on the chemical cues used by herbivorous insects in finding their host plant, the role of CO_2_ has generally been overlooked and this study addresses this knowledge gap. The data presented show that (1) in a 9 cm olfactometer the larvae are attracted to CO_2_ generated by carbonated water and that larvae silenced in one of the carbon dioxide receptor genes are no longer attracted. The behavior to three other compounds was not affected by the RNAi treatment; (2) the response to CO_2_ in the 9 cm olfactometer is CO_2_ – dose dependent, (3) silencing the receptor does not affect overall motility, but only behavioral responses to CO_2_. (4) in a 9cm olfactometer the larvae orient towards volatiles from roots of a maize seedling, and this is similar for WT and silenced larvae in the presence and absence of soda lime as a CO_2_ scrubber, however at 18 cm only WT larvae in the absence of the scrubber oriented towards the root volatiles. (5) in a tray assay the larvae appear to find the roots of plants more frequently with shorter distance from the roots and more orient towards high-fertilization treatment of plants than to low-fertilization treatment. The authors conclude that CO_2_ produced by maize roots is an important attractant for WCR larvae.

The fact that the study involves wildtype and RNAi silenced WCR larvae makes this a very interesting and innovative study and the outcome that CO_2_ is important for an insect herbivore in host-plant selection is new. Although I am impressed by the behavioral data presented for wildtype and silenced WCR larvae I have several important issues with the study.

1) Throughout the manuscript and in the title the authors conclude that the phenomena described can be ascribe to plant-derived CO_2_ – however there is no evidence for this at all. As mentioned in the Introduction, CO_2_ is a ubiquitous compound produced by almost all forms of life. The soil and especially the rhizosphere have abundant numbers of microbes that all produce CO_2_. The microbes in the rhizosphere occur in much higher densities than in the bulk soil as they are recruited by the roots. Thus microbial CO_2_ is also more abundant in the rhizosphere and this may be an important part of the explanation for the observed behavior. Thus, conclusion that the roots are the producer of the CO_2_ that attracts the WCR larvae (e.g. Results and also many other locations) cannot be drawn on the basis of the current study.

2) The statistics in Figure 3 cannot be correct and the asterisks seem to derive from a copy and paste process.

3) The fertilizer experiment lacks a proper control. Fertilization enhances nutrients in the soil and not only the plant but also the soil microbiome can exploit this nutrient source. As a consequence the effect of fertilizer may have been caused by enhanced activity and/or numbers of soil microbes producing higher levels of CO_2_. A control experiment using fertilizer in the absence of plants is needed to assess the effects of fertilizer on non-plant factors in the soil. Now the conclusion that fertilizer enhances CO_2_ production by plant roots is premature.

4) The Materials and methods section needs to become more informative. For instance, the method of CO_2_ quantification, a crucial element of the study is unclear: Subsection “Belowground olfactometer experiments with *DvvGr1*-, 462 *DvvGr2*-, *DvvGr3*-silenced WCR larvae“ states that this was done through GC-FID (really?) but does not tell at all how this was done. How was air sampled and transferred to the GC? This needs to be elaborated in detail as it is not straightforward. In the section on host location experiments two methods are mentioned for assessing CO_2_ levels: the use of a gas analyzer and the use of GC-FID but it is neither clear why two methods were used, nor how the air was sampled, nor which data derive from one or the other method. In the olfactometer, larval positions were recorded 1hour after release but how this was done is not clear. Neither is it clear to me whether in the 9 cm olfactometer the larvae can move over the full 9 cm or not. Same for the 18cm olfactometer. Insect behavior was observed for three minutes but it remains unclear how this was done. Same for how the trajectories were followed. Also it is not clear how the crawled distances were measured with ImageJ: were the paths filmed? The volatile-permeable root barrier (Results) has not been described in the Materials and methods section. These examples are non-exhaustive but rather an indicator that the methods section needs major improvement to allow others to repeat the experiments and to allow valuable evaluation of the data.

5) WCR larvae are soil dwelling and make their behavioral choices in the darkness in the soil. The behavioral assays have been done in Petri dishes and in olfactometers and in both cases the arena where the larvae were released does not seem to contain soil. At least, the picture in Figure 5B suggests that the middle part of the olfactometer where the larvae are introduced is empty. Were these bioassays carried out in the light or in darkness and if in darkness, how were the behavioral parameters quantified (Materials and methods)?

6) The use of soda lime to remove CO_2_ from the olfactometer setup is an interesting one but raises the question what the effect is on other volatiles. If CO_2_ levels were assessed through GC-FID such samples could also be used for GC-MS to assess the quantitative and qualitative effects of soda lime on other components of the volatile blend. The data in Figure 5B show that with a soda lime filter neither wildtype larvae nor silenced larvae move towards the volatiles from maize roots. The authors draw the conclusion that CO_2_ is needed for attraction to the roots, but an alternative explanation is that the soda lime also reduced the levels of other root volatiles with as consequence reduction below the level that leads to attraction at 18 cm distance. To assess which conclusion is best supporting the data the first step should be to analyze the volatiles with the method used already to assess CO_2_ levels.

Reviewer #2:

How insects (pests) locate host plants below ground is a highly relevant topic. The MS is very well written. The authors claim that they elucidated distance dependent mechanisms of host location and host quality assessment in WCR larvae by combining molecular, analytical, and behavioural techniques. However, only two findings endure scrutiny. (i) DvvGr2 is a CO_2_ receptor required for orientation in a CO_2_ gradient and (ii) CO_2_ attracts or repels WCR larvae in a concentration dependent manner. The Materials and method section suffers from a substantial information deficits impeding the reader to judge, to replicate, or to transfer the methods to other organisms.

1) The Introduction is very well written and gives an excellent overview on the state of the art. Nevertheless, there is potential to focus and shorten this section. The same applies to the Discussion, except for those sections that over-interpret results, see below and detailed comments.

2) The identification of putative CO_2_ receptors and selective silencing via RNAi appears to be performed in a competent way. But other reviewers certainly have more expertise in that area.

3) Synthetic CO_2_ has been applied (carbonated water, enriched air) and CO_2_ concentrations have been measured in divergent ways (GC-FID, IR-absorption). This apparently came with extreme SD values, which combined with low replicate numbers cast doubt on the results reported in Figure 2 (functional CO_2_ receptor assessment). As far as DvvGr2 is concerned these are compensated by the results reported in Figure 3 showing dose dependent responses to CO_2_ in WT but not in DvvGr2-silenced larvae. Unfortunately, this aspect is of limited novelty. Dose dependent responses to CO_2_ have been shown in many organisms, see publications referenced by the authors.

4) The authors do not differentiate between responses to volatile stimuli (in this set up: movement in a gradient) and responses to non-volatile stimuli (in this set up: arrest at the point of stimulus presentation). The experimental protocol does not allow for the diffusion of dissolved non-volatile stimuli. Responses to CO_2_ on the one hand and Fe(III)(DIMBOA)3 as well as a sugar blends on the other hand (Figure 2) therefore do not answer the question whether RNAi-silencing impairs attraction to other stimuli than CO_2_. However, the results presented in Figure 4 do overcome this shortcoming in showing that DvvGr2-silenced larvae respond to unknown maize root-derived stimuli. Removing the Fe(III)(DIMBOA)3 as well as the sugar blend experiments from the Results section would therefore be an option.

5) The conclusion suggesting CO_2_ as a long-distance attractant in a soil setting is not covered by the data presented. (i) The exp. reported in Figure 6A shows random movement and/or dispersal of the larvae in the absence of stimuli. (ii) CO_2_ concentrations above ambient have been ascertained in the zone adjacent to the maize plants only. (iii) Random movement as ascertained in Figure 6A leads to more frequent encounters of detectable CO_2_-gradients (or other chemical stimuli) in zone 2 the closer to this zone larvae have been released. (iv) Random movement combined with short distance attraction from zone 2 to zone 1 may therefore fully explain the higher proportion of WT compared to DvvGr2-silenced larvae ascertained within zone 1. The same line of arguments applies to the time course experiment (Figure 7—figure supplement 2). Thus, all results may be explained by other processes than long distance orientation to CO_2_ sources. Furthermore, observed differences between WT and Gr2-silenced larvae may well be due to interactions of CO_2_ gradients with other plant derived attractants at any level of sensory processing, since most of zone 2 lies within the range of effective attractiveness of plant volatiles as demonstrated in Figure 5A. However, results presented in Figure 5 can't be fully compared to those from Figure 6, since diffusion dynamics of both CO_2_ and other volatiles are different in a one-dimensional closed tube filled with another substrate compared to the open tray.

6) The conclusion suggesting that WCR larvae performed a choice between plants of different suitability based on CO_2_ emissions is not supported by the data presented. A choice based on comparative quality assessment implies that two stimuli are compared against each other. The CO_2_ concentrations displayed in Figure 7 and Figure S1 indicate that there was no gradient between the point of release (zone 3) and zone 4 adjacent to the low/medium fertiliser treatment, or ∆CO_2_ between zone 3 and zone 4 were in the range of no behavioural response as presented in Figure 3B. In contrast a significant gradient appears when comparing zone 3 and 2. Thus, WT larvae moved upwards the only available CO_2_ concentration gradient. This behaviour may have field relevance as high emitters become more prone to being discovered by WCR larvae. However, host quality assessment by the larvae appears a pretty premature conclusion.

7) The Materials and method section is incomplete in several aspects. Some instruments or providers are not specified. Operating conditions and sampling procedures are not described. No written descriptions are given for the olfactometer variants, the core bioassay device of the study. This extends to the substrate used, or references not providing the info claimed (e.g. GC-FID). Core information required to assess results and conclusions, to enable others to either replicate the study, or to apply the methods to other organisms is lacking.

Conclusion: Despite the topic being of outstanding interest to the scientific community and the reported results (DvvGr2 is a CO_2_ receptor; responses to the gas are dose dependent) meriting publication, the ascertained shortcomings of the manuscript as it currently stands prohibit publication in *eLife*.

Reviewer #3:

In this study Arce et al., functionally analyze three putative *Diabroticavirgiferavergifera* (WCR) CO_2_ receptor genes and use genetically silencing the expression of these genes to assess the role of CO_2_ as a cue in insect host finding behavior and host plant quality assessment. With a series of convincing combinations of molecular biology and bioassay experiments, the authors conclude that CO_2_ functions as a long-distance cue for host-searching WCR larvae. Moreover, WCR can use the CO_2_ cue to assess ost plant quality, which is closely linked to higher metabolic rates and thus higher belowground CO_2_ emissions.

This is a very complete study, excellently conducted and with solid and well supported conclusions. Overall the paper is very well written, and it is a pleasure to read through it. There are a small number of typos. However, overall, this is one of the best papers I have read in a while. I do not have any major concerns about the experimental procedures, the structure of the text or the interaction between results and conclusions.

[Editors’ note: further revisions were suggested prior to acceptance, as described below.]

Thank you for resubmitting your work entitled "Plant-associated CO_2_ mediates long-distance host location and foraging behaviour of a root herbivore" for further consideration by *eLife*. Your revised article has been evaluated by Meredith Schuman (Senior Editor) and a Reviewing Editor.

The manuscript has been thoroughly revised and includes additional experimental data as well as an effectively modified description of the methods.

The two reviewers agree that the manuscript has been significantly improved and they identify two major issues that need to be addressed before a final decision can be made.

1) The description of the GC-FID to measure CO_2_ (Materials and methods), which needs more details because a standard FID will not detect CO_2_.

2) The statistical analyses of many of the experiments are not appropriate and are in need of adjustment.

Reviewer #1:

1) This study's topic is very interesting, addressing the effect of carbon dioxide on the behavior of a root-feeding insect. Addressing the effect of carbon dioxide is novel and the approach taken by knocking down the carbon dioxide receptor gene of the insects is elegant. The experiments address different levels of scale and substrate and the data look very interesting. The manuscript is a revision of a previous version and it has greatly improved with more detailed descriptions of the methods and with additional CO_2_ measurements and root volatile quantification. This supports the understanding of the experiments carried out.

2) Although the data look very interesting, my main concern with the current version of the manuscript is the statistical analyses. Data for distributions of larvae over two arms of an olfactometer or five zones in a tray are dependent data and cannot be analyzed by ANOVA. Moreover, in some cases the statistical data in Supplementary file 1 conflict with the statistical data presented in the figures. The statistical analyses should be revised to allow a robust analysis of the data presented.

Statistics:

3) Results: why an ANOVA with multiple comparison test? This should be a simple two-sample t-test without multiple comparison test (what multiple comparison would be available anyway?).

4) Materials and methods: same comment as for above.

5) Results: the choice data are dependent data (a larva can either choose the treatment or the control) and so an ANOVA is not the right test. Moreover it is unclear why a two-way ANOVA was used followed by post hoc tests: the asterisks only seem to indicate whether the choice for treatment versus control differs from a 50:50 distribution.

6) Results: The asterisks next to the bars suggest to me that these relate to whether the larvae used for that bar had a distribution over the two odor sources that is significant from 50:50. But, how can this be analysed by a three-way ANOVA? There were three olfactometers, each with 6 larvae and the % larvae to either arm was recorded. So for each bar in the graph there are 3 datapoints of % larvae to CO_2_-enriched versus % of larvae to control (the two %% being dependent data).

7) Supplementary file 1 does not help me to understand the statistics either: here for Figure 4B the table tells me that for a 3-way ANOVA there is no main effect of Silencing construct (P=0.77; so the bars for the WT and silenced larvae do not differ overall) and not main effect of CO_2_ dose (P=1.00; so no overall effect of CO_2_ dose on larval behavior, no interaction of Silencing construct with CO_2_ dose (P=0.999; so the responses of WT and silenced larvae do no differ for the different CO_2_ doses). It is not clear from Supplementary file 1 what "treatment" is; it is not significant as main effect (P=0.25) but there is a significant interaction between Treatment and CO_2_ dose (P<0.001, indicating that the responses to CO_2_ doses are different from different treatments). The significant 3-way interaction between Silencing construct and treatment and CO_2_ dose is unclear to me (and 3-way interactions are difficult to interpret anyway, but it further adds to my confusion about the meaning of Sc and T).

8) Results: In the rebuttal the authors write " Note that the differences in CO_2_ concentrations between both sides of the olfactometers are significantly different at each concentration at a p-value<0.001; hence the many asterisks." which further suggests that a t-test was done for the two CO_2_ concentrations per test.

9) Results: see comment for point 3. Supplementary file 1 further suggests that a t-test was done.

10) The data in Figure 5C disagree with the data in Supplementary file1: Supplementary file 1 tells that the P value for crawled distance analysis is P=0.635, whereas the figure labels these data with ***, indicating that P<0.001.

11) Results: same comment as for point 6.

Here we also conducted GLM with binomial distribution followed by analysis of deviance and by FDR-corrected Least Square Means post hoc tests. We included four factors in the analyses -larval silencing, olfactometer arm size, presence of CO_2_ scrubber, and olfactometer side- but indicate the differences in larval choice within each silencing/arm size/CO_2_ scrubber combinations as these comparisons are more relevant to tests our hypothesis.

12) Results: The data for the larvae recovered at different distances are categorical data that are dependent: a larva found back in zone 1 cannot be found back in any of the other zones. An AOVA is not appropriate to analyse these data. Also the information in Supplementary file 1 does not provide information on the use of a categorical data analysis. The text further suggests that the data were analyzed per zone through an ANOVA which is not appropriate for these data.

13) Results: See earlier comments on the inappropriateness of an ANOVA because the data for zone 1 and zone 5 are dependent data and for a post-hoc test because no treatments are compared.

Reviewer #2:

How root feeding insects orient in the soil towards their host plants and the contribution of root-associated CO_2_ as well as other root-derived chemical stimuli to this process is still a matter of debate and in many aspects an open research question. In this study Arce et al. identify among three candidates one molecular receptor for CO_2_ perception in larvae of the western corn root worm (WCR). Through a series of behavioural experiments employing the RNAi knockdown technique they demonstrate that this single receptor is both essential and sufficient to mediate behavioural responses to CO_2_ in a dose dependent manner from attraction to repellence form low to high CO_2_ concentrations, respectively. Further experiments show that, while other root derived stimuli attract WCR larvae on a short range irrespective of the CO_2_ receptor being silenced or not, CO_2_ unfolds its behavioural activity over longer distances. The authors also demonstrate that maize roots establish CO_2_ gradients in the soil with steeper gradients around highly fertilised plants, while WCR larvae grow better when feeding on the latter. By showing that WCR larvae move towards the source in all CO_2_ gradients encountered under field-relevant conditions, they discuss an enhanced probability for well-nourished maize plants, which emit more CO_2_ compared to plants under a poor fertiliser regime, to be found by one of their major insect herbivores.

Strengths

The manuscript is very well written and gives an excellent overview on the state of the art as introduction. The combination of molecular techniques and behavioural experiments provides novel and unique insights into the orientation of a root feeding insect below ground and the role of CO_2_ as an important cue in host location over long distances compared to other root-derived volatiles. By using the RNAi technique the authors reliably demonstrate that the targeted CO_2_ receptor is not required for the insect to respond to other cues, be they volatile or soluble compounds like sugars. The multitude of experimental techniques is well documented. The Discussion highlights as a striking difference to other species that a single molecular receptor mediates the full range of behavioural responses to CO_2_. It also carefully addresses different aspects and stages of host location from random walk with no available cue to upward orientation within a CO_2_ gradient at longer distances while other root-derived volatiles gain behavioural relevance at shorter distances even whit a silenced CO_2_ receptor. In summary these results are noteworthy with respect to the notorious difficulties of assessing insect behaviour below ground.

Weaknesses

By applying a number of different behavioural experiments including volatile and non-volatile stimuli in settings with and without substrate the authors have to some extend limited the informative value of their otherwise excellent and comprehensive work. Responses to volatile and non-volatile stimuli not only involve different sensory modalities, but also different physical processes below ground. While volatile stimuli may – depending on their valence – serve as attractants or repellents, non-volatile stimuli, which are applied at a single point in an arena with no established concentration gradient, may only arrest insects upon contact. Consequently, the latter stimuli contribute little if anything to the authors quest of understanding how host plants are not only located but also assessed from a distance through sensing of CO_2_ and interacting volatile stimuli. This interaction aspect is of special interest since, as stressed by the authors themselves, CO_2_ is emitted by almost every respiring soil organism and therefore a cue of limited reliability.

Abstract – please reword to: „…is specifically required for dose-dependent responses.… but silencing does not impair responses to other cues, search behaviour or motility". Rationale: This is what your impressing set of experiments has shown. Still, whether CO_2_ “influences" responses to some chemical stimuli or vice versa remains another question. E.g., how would it interact with methyl anthranilate etc.

Abstract: add “cue” after location

Results: Please reword; unless a gradient of non-volatile cues like sugars or DIMBOA is comprehensibly established in the substrate these cues should not be coined „attractants". In your experiment these cues are exposed to the larvae on a piece of filter paper on a moist substrate. That means no or neglectable diffusion into the substrate. Obviously, these compounds arrest the larvae upon contact, while the control stimulus does not. Please also check the rest of the manuscript with respect to this issue.

Results and Figure 3: Changing the sequence within the Figure would help readers to quickly grasp that (D), (E), and (F) have been performed in the petri-dish experiment instead of the olfactometer. Please place the petri dish at position D, D to E, E to F etc.

Results: place „(D)" before ‚Mean…' to consistently introduce to the different parts of the figure/caption as with (A), (B) etc.

Figure 6: Negative values on the x-axis have shadowy white squares in-between. Please remove for the print version.

Materials and methods: To the best of my knowledge, a conventional FID does not detect CO_2_. However, with some adaptation of the instrument, e.g. a methaniser chamber located between the column outlet and the FID, CO_2_ detection and quantitation becomes feasible. Please provide respective info and very briefly outline how the instrument has been calibrated. Otherwise readers my wonder.…

Materials and methods: were instead of „where" and suggestion for rewording:.… were found at a distance of 1 cm or less from the.…

Legend to Figure S1F: Check assignment; it doesn't seem to fit to Figure 6 of the main text.

---

## [Author Response]

[Editors’ note: the authors resubmitted a revised version of the paper for consideration. What follows is the authors’ response to the first round of review.]

Reviewer #1:This manuscript has an interesting topic, presenting a study on the effect of CO_2_ on the behavior of a root-feeding insect, larvae of the western corn rootworm. Despite ample research on the chemical cues used by herbivorous insects in finding their host plant, the role of CO_2_ has generally been overlooked and this study addresses this knowledge gap. The data presented show that (1) in a 9 cm olfactometer the larvae are attracted to CO_2_ generated by carbonated water and that larvae silenced in one of the carbon dioxide receptor genes are no longer attracted. The behavior to three other compounds was not affected by the RNAi treatment; (2) the response to CO_2_ in the 9 cm olfactometer is CO_2_ – dose dependent, (3) silencing the receptor does not affect overall motility, but only behavioral responses to CO_2_. (4) in a 9cm olfactometer the larvae orient towards volatiles from roots of a maize seedling and this is similar for WT and silenced larvae in the presence and absence of soda lime as a CO_2_ scrubber, however at 18 cm only WT larvae in the absence of the scrubber oriented towards the root volatiles. (5) in a tray assay the larvae appear to find the roots of plants more frequently with shorter distance from the roots and more orient towards high-fertilization treatment of plants than to low-fertilizatipn treatment. The authors conclude that CO_2_ produced by maize roots is an important attractant for WCR larvae.The fact that the study involves wildtype and RNAi silenced WCR larvae makes this a very interesting and innovative study and the outcome that CO_2_ is important for an insect herbivore in host-plant selection is new. Although I am impressed by the behavioral data presented for wildtype and silenced WCR larvae I have several important issues with the study.

We thank the reviewer for the positive opinion on our study and also for all the constructive comments and suggestions, which we have addressed in the new version of our manuscript as indicated below.

1) Throughout the manuscript and in the title the authors conclude that the phenomena described can be ascribe to plant-derived CO_2_ – however there is no evidence for this at all. As mentioned in the Introduction, CO_2_ is a ubiquitous compound produced by almost all forms of life. The soil and especially the rhizosphere have abundant numbers of microbes that all produce CO_2_. The microbes in the rhizosphere occur in much higher densities than in the bulk soil as they are recruited by the roots. Thus microbial CO_2_ is also more abundant in the rhizosphere and this may be an important part of the explanation for the observed behavior. Thus, conclusion that the roots are the producer of the CO_2_ that attracts the WCR larvae (e.g. Results and also many other locations) cannot be drawn on the basis of the current study.

The reviewer is right to point out that many different organisms produce CO_2_, including roots themselves, root-associated microbes as well as free-living soil microbes. To evaluate the contribution of the roots and their microbiome to CO_2_ gradients in the soil, we now measured CO_2_ at different distances from maize plants before and after removing the seedlings from the soil (new Figure 1). We observe a strong CO_2_ gradient from the roots extending up to 32 cm into the soil. This gradient disappears rapidly when the plants are removed, and the difference can be fully explained by the CO_2_ that is released by washed roots. Thus, the presence of plants is tightly associated with higher CO_2_ levels in time and space, and can thus serve as a foraging cue for root-feeding insects. Note that we presently cannot distinguish between CO_2_ from root and root-associated microbial respiration. This question is beyond the scope of our study and not directly relevant to our research question, as the exact source of the CO_2_ does not matter for a root herbivore during long-distance host location. We now refer to “plant-associated CO_2_” in the manuscript title and throughout the text and mention that the CO_2_ is likely coming from both plant and microbial respiration.

2) The statistics in Figure 3 cannot be correct and the asterisks seem to derive from a copy and paste process.

We apologize for the confusion. Two statistical analyses were conducted in this case. The first one to compare the CO_2_ concentrations on each side of the olfactometers, and the second one to compare insect preferences. The asterisks indicating significant differences were placed next to the CO_2_ concentration values and next to the larval choice values, respectively. We modified the figure and mention this aspect in the figure legend. Note that the differences in CO_2_ concentrations between both sides of the olfactometers are significantly different at each concentration at a p-value<0.001; hence the many asterisks.

3) The fertilizer experiment lacks a proper control. Fertilization enhances nutrients in the soil and not only the plant but also the soil microbiome can exploit this nutrient source. As a consequence the effect of fertilizer may have been caused by enhanced activity and/or numbers of soil microbes producing higher levels of CO_2_. A control experiment using fertilizer in the absence of plants is needed to assess the effects of fertilizer on non-plant factors in the soil. Now the conclusion that fertilizer enhances CO_2_ production by plant roots is premature.

We thank the reviewer for pointing out this aspect. As described in the methods section, we first grew plants under different fertilization regimes. We then unearthed, washed, and replanted the plants into new soil trays with equal nutrient levels to avoid direct effects of the fertilizer treatment. Our setup allows for the conclusion that the higher CO_2_ concentrations are due to the effects of the fertilizer on the plant (and possibly its associated microbiome) rather than on the soil. We now clarify this in the paper.

4) The Materials and methods section needs to become more informative. For instance, the method of CO_2_ quantification, a crucial element of the study is unclear: Subsection “Belowground olfactometer experiments with DvvGr1-, 462 DvvGr2-, DvvGr3-silenced WCR larvae“ states that this was done through GC-FID (really?) but does not tell at all how this was done. How was air sampled and transferred to the GC? This needs to be elaborated in detail as it is not straightforward. In the section on host location experiments two methods are mentioned for assessing CO_2_ levels: the use of a gas analyzer and the use of GC-FID but it is neither clear why two methods were used, nor how the air was sampled, nor which data derive from one or the other method. In the olfactometer, larval positions were recorded 1hour after release but how this was done is not clear. Neither is it clear to me whether in the 9 cm olfactometer the larvae can move over the full 9 cm or not. Same for the 18cm olfactometer. Insect behavior was observed for three minutes but it remains unclear how this was done. Same for how the trajectories were followed. Also it is not clear how the crawled distances were measured with ImageJ: were the paths filmed? The volatile-permeable root barrier (Results) has not been described in the Materials and methods section. These examples are non-exhaustive but rather an indicator that the Materials and methods section needs major improvement to allow others to repeat the experiments and to allow valuable evaluation of the data.

We thank the reviewer for pointing this out. Reviewer # 2 also agrees that we fall short in describing some of our experimental approaches. We have reworked the Materials and methods section entirely and explain our experimental approaches in much more detail.

5) WCR larvae are soil dwelling and make their behavioral choices in the darkness in the soil. The behavioral assays have been done in Petri dishes and in olfactometers and in both cases the arena where the larvae were released does not seem to contain soil. At least, the picture in Figure 5B suggests that the middle part of the olfactometer where the larvae are introduced is empty. Were these bioassays carried out in the light or in darkness and if in darkness, how were the behavioral parameters quantified (Materials and methods)?

Thank you for this question. Behavioral experiments in Petri plates and olfactometers that did not involve plants were conducted in a dark room. As insects are insensitive to red light, red light headlamps were used to evaluate larval behavior. Experiments with olfactometers that involved plants were covered with aluminium to reduce light disturbance to the insects. This aspect is now included in the Materials and methods section.

6) The use of soda lime to remove CO_2_ from the olfactometer setup is an interesting one, but raises the question what the effect is on other volatiles. If CO_2_ levels were assessed through GC-FID such samples could also be used for GC-MS to assess the quantitative and qualitative effects of soda lime on other components of the volatile blend. The data in Figure 5B show that with a soda lime filter neither wildtype larvae nor silenced larvae move towards the volatiles from maize roots. The authors draw the conclusion that CO_2_ is needed for attraction to the roots, but an alternative explanation is that the soda lime also reduced the levels of other root volatiles with as consequence reduction below the level that leads to attraction at 18 cm distance. To assess which conclusion is best supporting the data the first step should be to analyze the volatiles with the method used already to assess CO_2_ levels.

We thank the reviewer for pointing out this aspect. We think it is important to consider this experiment in its entirety, including the short arm olfactometers. What we find with the short arm olfactometers is that the larvae are attracted to plant volatiles independently of their capacity to perceive CO_2_ or the presence of soda lime. Thus, there must be attractive root volatiles other than CO_2_ that diffuse into the olfactometer, independently of the presence of soda lime. To further validate the use of the soda lime, we now measured its impact of root volatile diffusion in a separate experiment, as suggested by the reviewer. For this, we transplanted maize seedlings into sand-filled spherical glass pots, delivered pure and humified air through the pots via an inlet port, and pulled it out through an additional port, in the opposite side, that was outfitted or not with soda lime. Root volatiles were trapped on Porapak filters. Note that maize roots release volatiles at extremely low levels, which make them hard to trap and detect in vivo. Therefore, we stimulated volatile release by wounding the roots of the seedlings prior to the experiment. Typical chromatograms are shown in Figure 6—figure supplement 2A, and relative quantifications of detected maize root volatiles are shown in Figure 6—figure supplement 2B. As expected, we do not observe any measurable impact of soda lime on root volatiles other than CO_2_. Taken together, these experiments provide strong experimental support for the hypothesis that CO_2_ is required for host attraction at longer distances.

Reviewer #2:How insects (pests) locate host plants below ground is a highly relevant topic. The MS is very well written. The authors claim that they elucidated distance dependent mechanisms of host location and host quality assessment in WCR larvae by combining molecular, analytical, and behavioural techniques. However, only two findings endure scrutiny. (i) DvvGr2 is a CO_2_ receptor required for orientation in a CO_2_ gradient and (ii) CO_2_ attracts or repels WCR larvae in a concentration dependent manner.

We are thankful to the reviewer for the extensive and thoughtful review on our study. We have now performed several additional experiments to support our conclusion that CO_2_ serves as a host attraction cue not a short distance (9 cm), but at longer distance (18 cm). Please also see our detailed responses below.

The Materials and method section suffers from a substantial information deficits impeding the reader to judge, to replicate, or to transfer the methods to other organisms.

We reworked the Materials and methods section to explain our experimental approaches and clear up misunderstandings.

1) The Introduction is very well written and gives an excellent overview on the state of the art. Nevertheless, there is potential to focus and shorten this section.

We thank the reviewer for this suggestion. Unless there are space constraints, we would prefer to keep the current length of the Introduction to provide a comprehensive overview of the state of the art.

The same applies to the discussion, except for those sections that over-interpret results, see below and detailed comments.

Along the same lines as with the introduction, we prefer to keep the current length of the Discussion, unless there are space constraints that would require shortening. We have amended the Discussion regarding the interpretation of our results (see detailed comments below).

2) The identification of putative CO_2_ receptors and selective silencing via RNAi appears to be performed in a competent way. But other reviewers certainly have more expertise in that area.

We thank the reviewer for this positive assessment.

3) Synthetic CO_2_ has been applied (carbonated water, enriched air) and CO_2_ concentrations have been measured in divergent ways (GC-FID, IR-absorption). This apparently came with extreme SD values, which combined with low replicate numbers cast doubt on the results reported in Figure 2 (functional CO_2_ receptor assessment). As far as DvvGr2 is concerned these are compensated by the results reported in Figure 3 showing dose dependent responses to CO_2_ in WT but not in DvvGr2-silenced larvae. Unfortunately, this aspect is of limited novelty. Dose dependent responses to CO_2_ have been shown in many organisms, see publications referenced by the authors.

We thank the reviewer for this comment. We gather that the reviewer is satisfied with the evidence we present for the involvement of *DvvGr2* in dose-dependent CO_2_ responsiveness. We agree that such responses have been shown in other systems. However, these experiments are necessary to set the stage for the following experiments which explore the impact of plant-associated CO_2_ on host attraction. To the best of our knowledge, this is the first time that a molecular manipulative approach is used to assess the role of plant-associated CO_2_ in plant-herbivore interactions as well as root-environment interactions in general.

4) The authors do not differentiate between responses to volatile stimuli (in this set up: movement in a gradient) and responses to non-volatile stimuli (in this set up: arrest at the point of stimulus presentation). The experimental protocol does not allow for the diffusion of dissolved non-volatile stimuli. Responses to CO_2_ on the one hand and Fe(III)(DIMBOA)3 as well as a sugar blends on the other hand (Figure 2) therefore do not answer the question whether RNAi-silencing impairs attraction to other stimuli than CO_2_. However, the results presented in Figure 4 do overcome this shortcoming in showing that DvvGr2-silenced larvae respond to unknown maize root-derived stimuli. Removing the Fe(III)(DIMBOA)3 as well as the sugar blend experiments from the Results section would therefore be an option.

We thank the reviewer for this suggestion. The choice experiments using Fe(III)(DIMBOA)_3_ and sugars allow us to test whether silencing *DvvGr2* impacts WCR responses to these non-volatile cues, and to certain extent to test for the additional impacts of *DvvGr2* on host location behavior, which is important, as pointed out by the reviewer, to interpret some of our results. We would therefore prefer to keep these datasets in the manuscript unless there are space constraints.

5) The conclusion suggesting CO_2_ as a long distance attractant in a soil setting is not covered by the data presented. (i) The exp. reported in Figure 6A shows random movement and/or dispersal of the larvae in the absence of stimuli. (ii) CO_2_ concentrations above ambient have been ascertained in the zone adjacent to the maize plants only. (iii) Random movement as ascertained in Figure 6A leads to more frequent encounters of detectable CO_2_-gradients (or other chemical stimuli) in zone 2 the closer to this zone larvae have been released. (iv) Random movement combined with short distance attraction from zone 2 to zone 1 may therefore fully explain the higher proportion of WT compared to DvvGr2-silenced larvae ascertained within zone 1. The same line of arguments applies to the time course experiment (Figure 7—figure supplement 2). Thus, all results may be explained by other processes than long distance orientation to CO_2_ sources.

Thank you very much for raising this important point. The clearest evidence for CO_2_ acting as a long-range host attractant comes from the olfactometer experiments, where we can clearly show that CO_2_ is important for attraction 18 cm away from the plant, but not 9 cm away from the plant. In this case, random movement does not explain the observed patterns, as the CO_2_ gradient extends to the release point of the olfactometers, and larvae typically make a choice immediately and then stay in the arm they decide to move into.

As the reviewer points out, random movement may indeed play a role in the soil arenas, with larvae moving randomly until they detect a CO_2_ gradient, and then moving up the gradient in a *DvvGr2*dependent manner. To gain further insight into this possibility, we conducted detailed CO_2_ measurements to determine the distribution of plant-associated CO_2_ in the soil at higher resolution (11 instead of 5 zones; new Figure 1). We see a gradient up to 32 cm away from the plant, after which CO_2_ concentrations level off and become statistically indistinguishable from CO_2_ levels of soil without plants. Thus, larvae released in zone 3 (the middle of the trays) and zone 2 (closer to the plants) will have a CO_2_ gradient available for host location. Larvae released in zone 4 and 5 likely need to move randomly first before coming across a CO_2_ gradient, even though we cannot exclude that they may also encounter localized CO_2_ gradients in these zones when encountering preferential gas-phase pathways. We now interpret our experiments in the light of these findings. Importantly, the data presented in Figure 7E-F and Figure 8 can be explained without the need for random larval movement.

Furthermore, observed differences between WT and Gr2-silenced larvae may well be due to interactions of CO_2_ gradients with other plant derived attractants at any level of sensory processing, since most of zone 2 lies within the range of effective attractiveness of plant volatiles as demonstrated in Figure 5A. However, results presented in Figure 5 can't be fully compared to those from Figure 6, since diffusion dynamics of both CO_2_ and other volatiles are different in a one-dimensional closed tube filled with another substrate compared to the open tray.

First, we have performed several experiments that show that *DvvGr2*-silencing does not impair sensory processing of other plant-derived behavioral cues. (1) The attractiveness of sugars and benzoxazinoids is unaffected by *DvvGr2*-silencing. (2) The repellent effect of the volatile methyl anthranilate is unaffected by *DvvGr2*-silencing. (3) The attraction of the western corn rootworm larvae to volatile blends in the absence of a CO_2_ gradient is unaffected by *DvvGr2*-silencing at a distance of <9 cm. We believe that these results provide strong evidence against the hypothesis that DvvGr2 modulates responses to other attractants.

Second, while the argument about other volatiles interacting with CO_2_ in a DvvGr2-specific manner cannot be fully excluded for short distances (Zone 2), it can be excluded for long distances, (Zone 3 and beyond). Therefore, we are confident that we are measuring true effects of CO_2_-mediated host location whenever releasing larvae further away from the plant.

6) The conclusion suggesting that WCR larvae performed a choice between plants of different suitability based on CO_2_ emissions is not supported by the data presented. A choice based on comparative quality assessment implies that two stimuli are compared against each other. The CO_2_ concentrations displayed in Figure 7 and Figure S1 indicate that there was no gradient between the point of release (zone 3) and zone 4 adjacent to the low/medium fertiliser treatment, or ∆CO_2_ between zone 3 and zone 4 were in the range of no behavioural response as presented in Figure 3B. In contrast a significant gradient appears when comparing zone 3 and 2. Thus, WT larvae moved upwards the only available CO_2_ concentration gradient. This behaviour may have field relevance as high emitters become more prone to being discovered by WCR larvae. However, host quality assessment by the larvae appears a pretty premature conclusion.

We agree with this point. What we show is that WCR larvae prefer well-fertilized plants, and that WCR grow better on those plants. We also show higher CO_2_ levels around well-fertilized plants, and that CO_2_-insensitive WCR larvae do not show any preference for differentially fertilized plants. We can therefore conclude that CO_2_ detection is important for WCR to locate well-fertilized plants. However, our setup is at this point not sufficient to conclude that this behaviour is due to host-quality assessment. We have toned down our conclusions accordingly.

7) The Materials and method section is incomplete in several aspects. Some instruments or providers are not specified. Operating conditions and sampling procedures are not described. No written descriptions are given for the olfactometer variants, the core bioassay device of the study. This extends to the substrate used, or references not providing the info claimed (e.g. GC-FID). Core information required to assess results and conclusions, to enable others to either replicate the study, or to apply the methods to other organisms is lacking.

We apologize for the lack of detailed explanations. We have reworked the Materials and methods section to improve this aspect.

Conclusion: Despite the topic being of outstanding interest to the scientific community and the reported results (DvvGr2 is a CO_2_ receptor; responses to the gas are dose dependent) meriting publication, the ascertained shortcomings of the MS as it currently stands prohibit publication in eLife.

We have done our best to address these shortcomings and hope that our changes will convince the reviewer that our work merits publication in *eLife*.

[Editors’ note: what follows is the authors’ response to the second round of review.]

The manuscript has been thoroughly revised and includes additional experimental data as well as an effectively modified description of the methods.The two reviewers agree that the manuscript has been significantly improved and they identify two major issues that need to be addressed before a final decision can be made.1) The description of the GC-FID to measure CO_2_ (Materials and methods), which needs more details because a standard FID will not detect CO_2_.2) The statistical analyses of many of the experiments are not appropriate and are in need of adjustment.

Thanks again for the evaluation of our manuscript. In this new version, we address all the concerns and include all the suggestions made by the reviewers as detailed in their comments below. The many comments about the statistics might have been caused by the inaccurate description of the tests we had conducted. We evaluated larval choices by Generalized Linear Models (GLM) with binomial distribution followed by analysis of deviance and by FDR-corrected Least Square Means post hoc tests. We now clearly mention which statistical tests were conducted for each experiment and updated the Supplementary file 1 that summarizes all the statistical results. Regarding the CO_2_ measurements using GC-FID. The reviewer is right. Our instrument is equipped with a methanizer that allows to detect and quantify CO_2_. We included these aspects in the Materials and methods section.

Reviewer #1:1) This study's topic is very interesting, addressing the effect of carbon dioxide on the behavior of a root-feeding insect. Addressing the effect of carbon dioxide is novel and the approach taken by knocking down the carbon dioxide receptor gene of the insects is elegant. The experiments address different levels of scale and substrate and the data look very interesting. The manuscript is a revision of a previous version and it has greatly improved with more detailed descriptions of the methods and with additional CO_2_ measurements and root volatile quantification. This supports the understanding of the experiments carried out.

We thank the reviewer for the positive impression of the current version of our study, which, thanks to the suggestions of the editors and reviewers, includes more data and a better description of the methods we used.

2) Although the data look very interesting, my main concern with the current version of the manuscript is the statistical analyses. Data for distributions of larvae over two arms of an olfactometer or five zones in a tray are dependent data and cannot be analyzed by ANOVA. Moreover, in some cases the statistical data in Supplementary file 1 conflict with the statistical data presented in the figures. The statistical analyses should be revised to allow a robust analysis of the data presented.

We thank the reviewer for raising this important point. Unfortunately, in some instances, we failed to properly describe the statistical tests we conducted and to accurately introduce this information in Supplementary file1. We have thoroughly revised the statistics and the Supplementary file 1 following your suggestions as detailed in all the comments below. In essence, we had evaluated the distribution of larvae by Generalized Linear Models (GLM) with binomial distribution followed by analysis of deviance and by FDR-corrected Least Square Means post hoc tests. Detailed responses to all the comments can be found below and are marked in blue in the manuscript.

Statistics:3) Results: why an ANOVA with multiple comparison test? This should be a simple two-sample t-test without multiple comparison test (what multiple comparison would be available anyway?).

The reviewer is right. We now statistically analysed the data by Student’s t test.

4) Materials and methods: same comment as for above.

In this case, as there are three groups, one-way ANOVA is the right test to conduct.

5) Results: the choice data are dependent data (a larva can either choose the treatment or the control) and so an ANOVA is not the right test. Moreover it is unclear why a two-way ANOVA was used followed by post hoc tests: the asterisks only seem to indicate whether the choice for treatment versus control differs from a 50:50 distribution.

The reviewer is right. Actually, we had analysed the data by Generalized Linear Models (GLM) with binomial distribution followed by analysis of deviance and by FDR-corrected Least Square Means post hoc tests. We included three factors in the analysis -larval silencing, CO_2_ dose, and olfactometer side-. The asterisks indicate significant differences in larval choices within each condition (larval silencing/CO_2_ dose combinations) as this are the most relevant comparisons to test our hypothesis.

6) Results: The asterisks next to the bars suggest to me that these relate to whether the larvae used for that bar had a distribution over the two odor sources that is significant from 50:50. But, how can this be analysed by a three-way ANOVA? There were three olfactometers, each with 6 larvae and the % larvae to either arm was recorded. So for each bar in the graph there are 3 datapoints of % larvae to CO_2_-enriched versus % of larvae to control (the two %% being dependent data).

We apologise for the confusion, as mentioned above, we conducted GLM with binomial distribution followed by analysis of deviance and by FDR-corrected Least Square Means post hoc tests. We included three factors in the analyses -larval silencing, CO_2_ dose, and olfactometer side- but only indicate the differences in larval choice within each larval silencing/CO_2_ dose combinations, which are the most relevant comparisons to test our hypothesis.

7) Supplementary file 1 does not help me to understand the statistics either: here for Figure 4B the table tells me that for a 3-way ANOVA there is no main effect of Silencing construct (P=0.77; so the bars for the WT and silenced larvae do not differ overall) and not main effect of CO_2_ dose (P=1.00; so no overall effect of CO_2_ dose on larval behavior, no interaction of Silencing construct with CO_2_ dose (P=0.999; so the responses of WT and silenced larvae do no differ for the different CO_2_ doses). It is not clear from Supplementary file 1 what "treatment" is; it is not significant as main effect (P=0.25) but there is a significant interaction between Treatment and CO_2_ dose (P<0.001, indicating that the responses to CO_2_ doses are different from different treatments). The significant 3-way interaction between Silencing construct and treatment and CO_2_ dose is unclear to me (and 3-way interactions are difficult to interpret anyway, but it further adds to my confusion about the meaning of Sc and T).

“Treatment” in this case means the olfactometer side where the larvae were recovered (either control, or CO_2_ enriched side). We have changed the Supplementary file 1 and refer to “olfactometer side” instead of treatment as it might be more intuitive for the readers. “Sc” and “T” mean “Silencing construct” and “Treatment”. We modified the Supplementary file 1 to clarify this aspect.

8) Results: In the rebuttal the authors write " Note that the differences in CO_2_ concentrations between both sides of the olfactometers are significantly different at each concentration at a p-value<0.001; hence the many asterisks." which further suggests that a t-test was done for the two CO_2_ concentrations per test.

As mentioned above, we had analysed this data including all the different factors in the model as it allows to statistically test their individual contributions but only indicate the differences in CO_2_ levels within each larval silencing/CO_2_ dose combinations, which are the most relevant comparisons to test our hypothesis.

9) Results: see comment for point 3. Supplementary file 1 further suggests that a t-test was done.

The reviewer is right. We now conducted Student’s *t* tests. We modified the Supplementary file 1 to reflect this aspect.

10) The data in Figure 5C disagree with the data in Supplementary file 1: Supplementary file 1 tells that the P value for crawled distance analysis is P=0.635, whereas the figure labels these data with ***, indicating that P<0.001.

We have revisited the statistical results of these data sets and found out that the given P values were switched, because we have changed the order of the panels. We modified the Supplementary file 1 to reflect this change.

11) Results: same comment as for point 6.

Here we also conducted GLM with binomial distribution followed by analysis of deviance and by FDR-corrected Least Square Means post hoc tests. We included four factors in the analyses -larval silencing, olfactometer arm size, presence of CO_2_ scrubber, and olfactometer side- but indicate the differences in larval choice within each silencing/arm size/CO_2_ scrubber combinations as these comparisons are more relevant to tests our hypothesis.

12) Results: The data for the larvae recovered at different distances are categorical data that are dependent: a larva found back in zone 1 cannot be found back in any of the other zones. An AOVA is not appropriate to analyse these data. Also the information in Supplementary file 1 does not provide information on the use of a categorical data analysis. The text further suggests that the data were analyzed per zone through an ANOVA which is not appropriate for these data.

As mentioned before, we conducted GLM with binomial distribution followed by analysis of deviance and by FDR-corrected Least Square Means post hoc tests to statistically evaluate this data. We updated the Supplementary file 1 to reflect this aspect.

13) Results: see earlier comments on the inappropriateness of an ANOVA because the data for zone 1 and zone 5 are dependent data and for a post-hoc test because no treatments are compared.

The reviewer is right to point out that non-volatile compounds such as sugars or Fe(III)(DIMBOA)_3_ are likely less relevant for host location at long distances. We tested these compounds only to evaluate whether silencing CO_2_ receptor could influence larval responses to other chemicals apart from CO_2_, which is not the case. We characterized the behavioural relevance of sugars and BXDs in more detail in two additional studies (Hu et al., 2018; Machado et al., 2021).

Reviewer #2:StrengthsThe manuscript is very well written and gives an excellent overview on the state of the art as introduction. The combination of molecular techniques and behavioural experiments provides novel and unique insights into the orientation of a root feeding insect below ground and the role of CO_2_ as an important cue in host location over long distances compared to other root-derived volatiles. By using the RNAi technique the authors reliably demonstrate that the targeted CO_2_ receptor is not required for the insect to respond to other cues, be they volatile or soluble compounds like sugars. The multitude of experimental techniques is well documented. The Discussion highlights as a striking difference to other species that a single molecular receptor mediates the full range of behavioural responses to CO_2_. It also carefully addresses different aspects and stages of host location from random walk with no available cue to upward orientation within a CO_2_ gradient at longer distances while other root-derived volatiles gain behavioural relevance at shorter distances even whit a silenced CO_2_ receptor. In summary these results are noteworthy with respect to the notorious difficulties of assessing insect behaviour below ground.

We thank the reviewer for the positive impression of our study and for highlighting its different strengths.

WeaknessesBy applying a number of different behavioural experiments including volatile and non-volatile stimuli in settings with and without substrate the authors have to some extend limited the informative value of their otherwise excellent and comprehensive work. Responses to volatile and non-volatile stimuli not only involve different sensory modalities, but also different physical processes below ground. While volatile stimuli may – depending on their valence – serve as attractants or repellents, non-volatile stimuli, which are applied at a single point in an arena with no established concentration gradient, may only arrest insects upon contact. Consequently, the latter stimuli contribute little if anything to the authors quest of understanding how host plants are not only located but also assessed from a distance through sensing of CO_2_ and interacting volatile stimuli. This interaction aspect is of special interest since, as stressed by the authors themselves, CO_2_ is emitted by almost every respiring soil organism and therefore a cue of limited reliability.

The reviewer is right to point out that non-volatile compounds such as sugars or Fe(III)(DIMBOA)_3_ are likely less relevant for host location at long distances. We tested these compounds only to evaluate whether silencing CO_2_ receptor could influence larval responses to other chemicals apart from CO_2_, which is not the case. We characterized the behavioural relevance of sugars and BXDs in more detail in two additional studies (Hu et al., 2018; Machado et al., 2021).

Abstract – please reword to: “…is specifically required for dose-dependent responses.… but silencing does not impair responses to other cues, search behaviour or motility2”. Rationale: This is what your impressing set of experiments has shown. Still, whether CO_2_ “influences” responses to some chemical stimuli or vice versa remains another question. E.g., how would it interact with methyl anthranilate etc.

We thank the reviewer for this suggestion. We modified the text accordingly.

Abstract: add “cue” after location

Corrected.

Results: Please reword; unless a gradient of non-volatile cues like sugars or DIMBOA is comprehensibly established in the substrate these cues should not be coined “attractants”. In your experiment these cues are exposed to the larvae on a piece of filter paper on a moist substrate. That means no or neglectable diffusion into the substrate. Obviously, these compounds arrest the larvae upon contact, while the control stimulus does not. Please also check the rest of the manuscript with respect to this issue.

Reviewer # 1 made a similar comment. We modify the manuscript accordingly.

Results and Figure 3: Changing the sequence within the Figure would help readers to quickly grasp that (D), (E), and (F) have been performed in the petri-dish experiment instead of the olfactometer. Please place the petri dish at position D, D to E, E to F etc.

We thank the reviewer for this suggestion. We modified the figure accordingly.

Results: place “(D)” before “Mean…” to consistently introduce to the different parts of the figure/caption as with (A), (B) etc.

Corrected.

Figure 6: Negative values on the x-axis have shadowy white squares in-between. Please remove for the print version.

We will provide high resolution figures upon acceptance.

Materials and methods: To the best of my knowledge, a conventional FID does not detect CO_2_. However, with some adaptation of the instrument, e.g. a methaniser chamber located between the column outlet and the FID, CO_2_ detection and quantitation becomes feasible. Please provide respective info and very briefly outline how the instrument has been calibrated. Otherwise readers my wonder.…

Thanks for pointing this important technical point out. The GC-FID is equipped with a methanizer. More details in this context are now given in the Materials and methods section.

Materials and methods: were instead of „where" and suggestion for rewording:.… were found at a distance of 1 cm or less from the.…

Thanks for the suggestion. We modified the text accordingly.

Legend to Figure S1F: Check assignment; it doesn't seem to fit to Figure 6 of the main text.

Indeed, it corresponds to Figure 7. We modified the figure legend accordingly.